# Towards Understanding Continual Factual Knowledge Acquisition of Language Models: From Theory to Algorithm

**Haoyu Wang** [1]  **Yifan Shang** [1]  **Zhongxiang Sun** [1]  **Weijie Yu** [2]  **Xiao Zhang** [1]  **Jun Xu** [1]

## Abstract

Continual Pre-Training (CPT) is essential for enabling Language Models (LMs) to integrate new knowledge without erasing old. While classical CPT techniques like data replay have become the standard paradigm, the mechanisms underlying how LMs acquire and retain facts over time, termed as continual Factual Knowledge Acquisition (cFKA), remain unclear. In this work, we present a theoretical framework that characterizes the training dynamics of cFKA using a single-layer Transformer, offering a unified explanation for the behavior of representative CPT methods. Our analysis reveals that regularization-based methods merely adjust the convergence rate of parameters without altering the inherent forgetting tendency, whereas data replay methods succeed in shifting convergence dynamics and stabilizing pretrained knowledge. Building on these insights, we propose a novel generative data replay approach, called **S**electing **T**okens via attenti**O**n **C**ontribution (STOC), which identifies influential factual snippets to guide replay data generation. Extensive experiments on both synthetic and real-world datasets validate our findings and demonstrate that STOC effectively enhances cFKA by mitigating catastrophic forgetting.

## 1. Introduction

Large language models (LLMs) have acquired substantial factual knowledge during open-domain Pre-Training (PT) (Lin et al.; Wang et al., 2023), yet they still require billions of tokens in Continual Pre-

Training (CPT) (Yang et al., 2025; Ke et al., 2023; Yıldız et al., 2024) to adapt to newly emerging knowledge or domain-specific downstream applications (Tu et al., 2025; Mo et al., 2025; Lai et al., 2024; Lee et al., 2025). For instance, CPT may incorporate high-quality legal corpora to improve performance in downstream legal-advice applications (Sun, 2023). Similar to other Continual Learning (CL) settings, while new knowledge can be learned, the original tends to be forgotten, referred to as *catastrophic forgetting* (Zheng et al., 2025; Zucchet et al.). With the increasing integration of LLMs into diverse industries, understanding the learning dynamics of continual Factual Knowledge Acquisition (cFKA) (Ou et al., 2025; Wang et al., 2025), i.e., how LMs memorize newly introduced facts together with the pretrained during CPT, is essential for developing advanced techniques (Wang et al., 2025).

Several theoretical analyses have been employed to characterize the training dynamics of LMs (Tian et al., 2023; Nichani et al., 2025; Ren & Sutherland, 2024). However, it still remains unclear how standard continual learning techniques affect cFKA dynamics and mitigate catastrophic forgetting, particularly when factual knowledge is embedded in complex textual expressions. In fact, every piece of fact can be conveyed through multiple formats, inducing complex interactions among each other during multi-stage training. Therefore, analyzing how LMs integrate such heterogeneous expressions into unified representations and memorization is a critical step toward understanding cFKA. See Appendix B for more discussions on related works.

To demystify the cFKA process, we formally characterize how the learnable parameters evolve for a simplified single-layer Transformer, whose modules are assumed to be tuned with different learning rates. Within this mathematical framework, we prove several key properties: (1) The factual knowledge is decomposed into frequency-based information and stored at the token level. (2) The attention score assigned to a specific token is governed by the diversity of its associated answer distribution. Since PT and CPT share the same Next Token Prediction objective, we use a synthetic PT task as a cross-setting validation for our theoretical predictions. Specifically, we validate the framework by examining whether its predicted behaviors emerge in PT,

[1]Gaoling School of Artificial Intelligence, Renmin University of China, Beijing, China [2]School of Artificial Intelligence and Data Science, University of International Business and Economics, Beijing, China. Correspondence to: Xiao Zhang <zhangx89@ruc.edu.cn>.

*Proceedings of the $43^{rd}$ International Conference on Machine Learning*, Seoul, South Korea. PMLR 306, 2026. Copyright 2026 by the author(s).

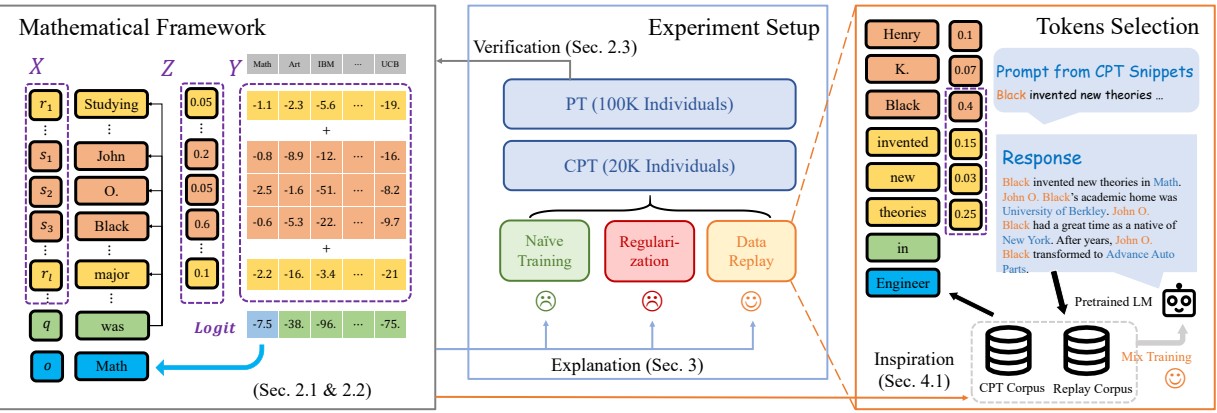

*Figure 1.* The overall structure of the paper. In Section 2, we present our mathematical framework and main theoretical results, whose effectiveness is verified during controlled PT. After that, we analyze popular continual learning methods in Section 3. Inspired by these findings, we propose a new generative data replay method in Section 4. Overall Notations are provided in Table 5.

thereby providing support for applying the framework to subsequent cFKA analysis. This validation further shows that the behaviors predicted by the simplified model also arise in modern LMs, strengthening the explanatory power of our analysis. Figure 1 illustrates the overall roadmap and logical structure of this paper.

Building on these results, we provide a new perspective on the mechanisms by which widely used CPT techniques affect cFKA. On the one hand, we demonstrate that regularization-based methods can alter merely the convergence rate rather than the convergence point for the cFKA dynamics. Thus, they generally fail in altering the inherent forgetting tendency in the cFKA scenario. On the other hand, we find that even a small proportion of replay data can substantially mitigate catastrophic forgetting. Beyond shifting the convergence point, data replay amplifies the oscillation amplitude near convergence, thereby improving the retention of pretrained knowledge. We further verify these findings in multi-layer LMs using the synthetic task, and the observation aligns with our prediction.

Furthermore, we propose a new generative data replay method called **S**electing **T**okens via attenti**O**n **C**ontribution (STOC). It is motivated by the need to examine the token-level overlap between new and existing knowledge. By calibrating the answer distributions associated with these overlapping tokens, STOC balances the integration of new knowledge with the preservation of pretrained knowledge. Specifically, STOC selects segments from CPT data based on token-level attention contributions and uses them as prompts for replay generation, eliciting responses from the pretrained LM that contain pretrained factual knowledge. In two representative CPT scenarios, experiments on both synthetic and real-world datasets indicate that STOC effectively mitigates catastrophic forgetting to improve overall performance.

## 2. Learning Dynamics of FKA

In this section, we investigate how learnable parameters evolve during training to characterize the memorization behaviors of LMs. Under assumptions of structured inputs, a single-layer LM with linear attention, and SGD training with different updating rates, we build a tractable framework and theoretically estimate the training dynamics of learnable parameters. Before applying this framework to continual Factual Knowledge Acquisition (cFKA) in Section 3, we first validate whether the derived dynamics are reflected under a controlled Pre-Training (PT) setting. This step avoids a circularity concern: if the theory were validated only in the Continual Pre-Training (CPT) phase, the validation would rely on the same setting that motivates our later cFKA analysis and STOC design. Moreover, PT provides a cleaner environment by removing confounding effects from pretrained initialization. The results show that, our framework explains several empirical phenomena in FKA, including how data augmentation enables LMs to generalize across different textual formats (Allen-Zhu & Li, 2024). These results can be observed not only in the toy single-layer LM but also in modern LMs, providing empirical support for the potential that the theoretical results can be extended to practical algorithms later in Section 4.

### 2.1. Input Structure and Model Architecture

**Input Structure** We suppose all the factual knowledge can be represented by "(subject, relation, object)" triples (Paulheim, 2016), which is commonly used in both theoretical (Nichani et al., 2025) and empirical (Allen-Zhu & Li, 2024; Zheng et al., 2025) works. Without loss of generality, every subject/relation/object is considered drawn from certain candidate sets. Particularly, we assume a causal graph where the random variable object is the ancestor of the subject and relation. For example, one born in China is more

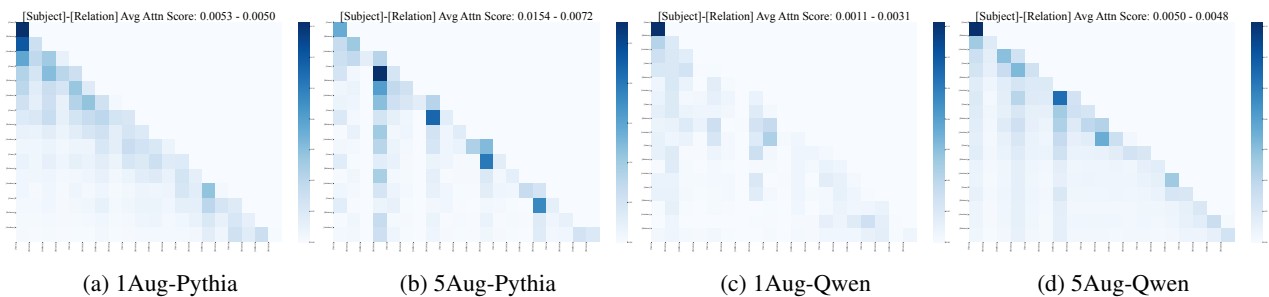

*Figure 2.* The attention assignment over different types of tokens in the test biography of the example individual. The assigned attention scores are averaged across tokens within each category (subject or relation). See detailed token-level attention score in Fig. 5 and 7.

likely to be named "Chen" and the relationship between an individual and a company is most likely an employment relationship. For convenience, we regard each subject can be tokenized into $K$ tokens, each object corresponds to a unique token. For example in Figure 1(left), a person's full name can be divided into first name, middle name, last name token, and his major can be "Math". Besides, we suppose the relation can be conveyed by templates with $L$ tokens, e.g., the relation about major can be "Studying until midnight every day, [Name]'s major was [Major]". Also, we suppose there is always a unique query token $x_{L+1} = q$ before $x_{L+2} = o$ such as "was", which acts as the pivot to calculate attention scores for each token.

**Model Architecture**  We employ a simplified one-layer transformer as the knowledge learner, such simplification is common in previous works (Tian et al., 2023; Ahn et al., 2023; Nichani et al., 2024; Zhang et al., 2024b; Nichani et al., 2025). First, we set $D$ as the vocabulary size and let $x_l \in [D]$ be discrete token, $\boldsymbol{x}_l = \boldsymbol{e}_{x_l} \in \mathbb{R}^D$ be one-hot vector, $\boldsymbol{X} = [\boldsymbol{x}_1, \boldsymbol{x}_2, \ldots, \boldsymbol{x}_L]^\top \in \mathbb{R}^{L \times D}$ be the input matrix. Next, word embedding $\boldsymbol{E} \in \mathbb{R}^{D \times d}$ is applied. Then,there is a single-head attention module parameterized by $\boldsymbol{W}_Q, \boldsymbol{W}_K, \boldsymbol{W}_V, \boldsymbol{W}_O \in \mathbb{R}^{d \times d}$, outputing

$$\boldsymbol{a} = \sigma\left(\boldsymbol{X}\frac{\boldsymbol{E}\boldsymbol{W}_K\boldsymbol{W}_Q^\top\boldsymbol{E}^\top}{\sqrt{d}}\boldsymbol{x}_{L+1}\right), \boldsymbol{h} = \boldsymbol{W}_O\boldsymbol{W}_V^\top\boldsymbol{E}^\top\boldsymbol{X}^\top\boldsymbol{a}.$$

Finally, we assume tied unembedding matrix $\boldsymbol{E}^\top$, yielding

$$\text{logit}(o \mid \boldsymbol{X}) = \boldsymbol{x}_o^\top\boldsymbol{E}\boldsymbol{h}, \hat{p}(\cdot|\boldsymbol{X}) = \text{Softmax}(\text{logit}(\cdot \mid \boldsymbol{X})).$$

Equivalently, the whole model can be re-parameterized as

$$\boldsymbol{Y} := \boldsymbol{E}\boldsymbol{W}_O\boldsymbol{W}_V^\top\boldsymbol{E}^\top \in \mathbb{R}^{D \times D},$$
$$\boldsymbol{Z} := \boldsymbol{E}\boldsymbol{W}_K\boldsymbol{W}_Q^\top\boldsymbol{E}^\top/\sqrt{d} \in \mathbb{R}^{D \times D}.$$

We remark $y_{o,s} = Y_{o,s}$ represents how much token $s$ supports the object $o$ through $\text{logit}(o \mid \boldsymbol{X})$, and $a_s = \sigma(Z_{s,q})$ determines the non-normalized attention score for token $s$. In the toy example of Figure 1, token "John" contributes to

the logit of token "Math" by $y_{o,r_1} = -0.8$ with attention weight $z_{r_1} = 0.2$. Intuitively, $\boldsymbol{Y}$ can be understood as simplified FFN modules storing the acquired knowledge in the modern LMs, which maps subject tokens and relation tokens to object tokens (Geva et al., 2021). $\boldsymbol{Z}$ can be understood as Attention parameters in Transformers (Vaswani et al., 2017), playing a critical role in moving information to the query token for prediction (Sun et al.).

**Training Method**  Cross-Entropy loss is adopted:

$$\mathcal{L} = -\text{logit}(x_{T+2} \mid \boldsymbol{X}) + \log \sum_o \exp\left(\text{logit}(x_o \mid \boldsymbol{X})\right).$$

As many previous works do (Tian et al., 2023; Li et al., 2023; Chen et al., 2024), we assume the learning rate satisfies $\eta_Y \gg \eta_Z$, so that we consider $z$ to be static when analyzing $\boldsymbol{Y}$'s dynamics. Based on these assumptions, we can derive the approximate evolving dynamics of the parameters, thereby characterizing the model's factual learning behavior. Detailed proof is in the Appendix E.

**Remark**  We acknowledge that there are many simplifications in the theoretical analysis for theoretical tractability and clarity. Despite the simplicity, it yields compelling mechanistic explanations for the behaviors of complicated modern LMs. More discussion about potential extensions such as generalized knowledge format, model architechure and limitations is provided in Appendix C and D.

## 2.2. Training Dynamics Induced by SGD

**$\boldsymbol{Y}$'s Dynamics**  Assuming $\boldsymbol{Z}$ remains unchanged, we first analyze the convergence of $\boldsymbol{Y}$. Notice loss function $\mathcal{L}$ is convex with respect to $\boldsymbol{Y}$, we therefore reveal a reference state $\boldsymbol{U} \in \mathbb{R}^{D \times D}$ and corresponding "token contribution"

$$\boldsymbol{U} = \sum_o \sum_s \frac{1}{a_s}\left[\ln(\Pr(s|o) + \frac{1}{L}\ln\Pr(o)\right] \cdot \boldsymbol{x}_o\boldsymbol{x}_s^\top,$$

which can be verified to achieve optimal Bayesian prediction. We then prove that the distance between $\boldsymbol{Y}$ and $\boldsymbol{U}$ consistently decreases to show the learning dynamics.

*Table 1.* Performance of the LMs with different augmentation strategies. 5/1/P-Aug represents that one individual corresponds to 5/1/Possion($\lambda = 5$) biographies. Green cell indicates that there is no significant generalization gap, while pink cell indicates a significant generalization gap conversely.

| Aug | Pythia-160m (Train) | | | Pythia-160m (Test) | | | Qwen2.5-0.5B(Train) | | | Qwen2.5-0.5B(Test) | | |
|---|---|---|---|---|---|---|---|---|---|---|---|---|
| | hFTA | sFTA | EM | hFTA | sFTA | EM | hFTA | sFTA | EM | hFTA | sFTA | EM |
| 5-Aug | 95.44 | 93.64 | 81.88 | 94.65 | 92.64 | 81.27 | 95.75 | 94.63 | 83.93 | 95.08 | 93.88 | 83.43 |
| 1-Aug | 95.82 | 95.27 | 8.85 | 15.42 | 14.43 | 2.71 | 96.15 | 95.76 | 51.43 | 13.80 | 12.98 | 2.94 |
| P-Aug | 95.46 | 93.52 | 81.30 | 94.37 | 91.77 | 78.06 | 95.99 | 95.43 | 83.73 | 95.25 | 94.49 | 83.18 |

**Theorem 1** (Dynamics of $\boldsymbol{Y}$). *Let $\boldsymbol{\xi}(t) := \boldsymbol{x}_{L+2}(t) - \mathrm{softmax}(\sum_s z_s \delta_s(t) \boldsymbol{u}_s)$ represents the perturbation term, and let error term after $t$ step updates be $\boldsymbol{e}_s(t) := \boldsymbol{y}_s(t) - \boldsymbol{u}_s$. Then, using 1st order Taylor expansion we have:*

$$\boldsymbol{e}_s(T) \approx \left[ \prod_{t=1}^{T} \left( \boldsymbol{I} - \eta_Y z_s \delta_s(t) \tilde{\boldsymbol{H}}(t) \right) \right] \boldsymbol{e}_s(0)$$
$$+ \sum_{t=1}^{T} \eta_y z_s \delta_s(t) \left[ \prod_{\tau=t+1}^{T} \left( \boldsymbol{I} - \eta_Y z_s \delta_s(\tau) \tilde{\boldsymbol{H}}(\tau) \right) \right] \boldsymbol{\xi}(t),$$

$$(1)$$

*where $\delta_s(t)$ is an indicator variable of token $s$, $\tilde{\boldsymbol{H}}(t) = \mathrm{diag}(\tilde{\boldsymbol{x}}(t)) - \tilde{\boldsymbol{x}}(t)\tilde{\boldsymbol{x}}(t)^\top$ is the Jacobian matrix, $\tilde{\boldsymbol{x}}(t) = \mathrm{softmax}\left(\sum_{s'} z_{s'} \delta_{s'}(t) \boldsymbol{u}_{s'}\right)$ is the optimal prediction.*

Eq. (1) indicates that throughout the gradient flow, $\boldsymbol{Y}$ gradually approaches $\boldsymbol{U}$ and ultimately oscillates around its vicinity, like a damped oscillator. The first term of exponential decay determines the training convergence rate by the largest eigenvalue $\lambda_{\max}^+(\tilde{\boldsymbol{H}})$. For an information-rich token $s$ like "John", $\tilde{\boldsymbol{x}}$ will be polarized and $\lambda_{\max}^+(\tilde{\boldsymbol{H}})$ will be small. Given the same optimization steps, $\boldsymbol{y}_s$ will exhibit a smaller error and achieve quick convergence, indicating that the LM learns factual knowledge of "John" fast. The second oscillation term remains at a fixed amplitude of $O(\lambda_{\min}^+(\tilde{\boldsymbol{H}}))$ as $\boldsymbol{\xi}(t)$ is bounded, constraining the range of $\boldsymbol{Y}$ and therefore determining the error bar at convergence. This term, naturally inherited from SGD training, can serve as a reminder to help LM remember training samples, especially those with low frequency. Also, if "John" rarely appears in the training corpus, then a bigger $\lambda_{\min}^+(\tilde{\boldsymbol{H}})$ will lead to better factual memory. In short, $\boldsymbol{Y}$ partitions knowledge into tokens and organizes them by associating each token with its co-occurring object tokens from a frequency perspective. Such token-level knowledge is aggregated through the attention module to provide a guide for object prediction.

> **Fact-to-Frequency Abstraction:** The factual knowledge is decomposed into frequency-based information and stored at the token level. For one certain token, the convergence rate and oscillation amplitude depend on the flatness of its associated tokens.

**$\boldsymbol{z}$'s Dynamics** We then start to analyze how $\boldsymbol{Z}$ changes in the training process. For the ease of qualitative analysis and following previous works (Dai et al., 2023; Yao et al., 2024), we here ignore the normalization term and set $\sigma$ as an identity mapping as a relaxation. Intuitively, LMs should learn to assign attention scores according to token importance. The following theorem indicates that such importance is measured through the $\ell_2$-norm of $\boldsymbol{y}_s$.

**Theorem 2** (Dynamics of $\boldsymbol{Z}$). *The following quantity remains constant throughout the training:*

$$\frac{d}{dt} \left[ \left( \frac{1}{\eta_Z} z_s \right)^2 - \sum_o \left( \frac{1}{\eta_Y} y_{o,s} \right)^2 \right] = 0. \qquad (2)$$

Theorem 2 reveals the relationship between the values of $z_s$ and $\boldsymbol{y}_s$. Neglecting the influence of the periodic perturbation and initialization term by setting $\boldsymbol{y}_s = \boldsymbol{u}_s$ for convenience, Eq. (2) specifies a pattern by which the converged model allocates attention scores through a certain Diversity Index (DI, (Simpson, 1949)). Let $C$ be a constant determined by initialization, the DI is defined as

$$\mathrm{DI}(\overline{\boldsymbol{x}}_s) \propto -\sqrt{\frac{\eta_Z}{\eta_Y}} \sqrt[4]{\sum_o \left[ \ln \Pr(s|o) + L^{-1} \ln \Pr(o) \right]^2} + C. \qquad (3)$$

For a token $s$ whose associations are more balanced and broadly distributed across objects, such as "the", it functions as a common token with low object-specific information content. In our definition, such tokens have a higher Diversity Index (DI), since they are associated with a greater variety of objects, and $\ln \Pr(s|o)$ is typically negative with a large absolute value. Accordingly, the model assigns lower attention scores to these less informative tokens. Therefore, our experimental results demonstrate that tokens with lower information content, namely higher DI, receive lower attention weights, which is fully consistent with the experiments in Section 2.3 and the main theoretical arguments of our paper. In all, we can describe the attention allocation as:

> **Diversity-Aware Attention Assignment:** The LM assigns attention scores according to the related-object diversity of each token. Information-poor tokens will receive lower attention scores.

## 2.3. Validation For the Analysis Framework

Through several bold assumptions, we have derived several properties of the LMs. In this section, we empirically validate these results during PT stage. Rather than restricting to the single-layer setting, we further examine whether these properties also emerge in modern multi-layer language models. As a representative example, we illustrate how data augmentation enables LMs to generalize across different knowledge formats. More complementary explanation of grokking (Zucchet et al.) is in Appendix H.1.

**Dataset**   To eliminate the overlap between PT's and CPT's knowledge and training samples, we first introduce a fully synthetic task for controlled experiments, which follows the setup in (Allen-Zhu & Li, 2024). The constructed `Biography` dataset contains 120,000 individuals, each characterized by five relations: birthday, birthplace, university, major, and company. One biography text can be generated by inserting an individual's name and attributes into pre-defined templates. It can be readily verified that such an input format aligns with the assumptions established earlier. In our quest to understand the role of data augmentation, we adopt three different settings for training biography construction: (1) one biography per individual; (2) five biographies per individual; and (3) the number of biographies per individual follows a Poisson distribution of $\lambda = 5$. At the same time, every individual is provided with another three biographies for evaluation, where individual names and templates are used as prompts to generate corresponding attributes. To examine the LMs' behavior of FKA during every stage, we then divide the dataset into the PT and CPT corpora at a 5:1 ratio with respect to the number of individuals. See detailed description, rationale discussion, and example demonstration in the Appendix C and F.

**Direct Validation**   We first directly check the relationship between token-level attention scores and the proposed DI in the single-layer setting across two attention variants: linear attention and softmax attention. The latter corresponds to the analysis of exponential attention presented in Appendix C. These two models are trained on the settings in

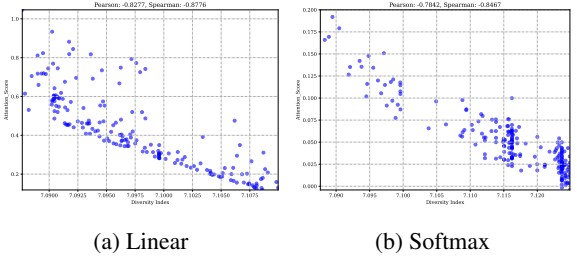

(a) Linear       (b) Softmax

*Figure 3.* Attention Score and the corresponding Diversity Index. The Correlation is significant across two attention.

Sec 2.2 where "Last Token Prediction" is adopted on the first object token. The empirical results are presented in Figure 3. Across both attention mechanisms, we observe a strong negative correlation between token attention scores and diversity index, with both Pearson and Spearman correlation coefficients below $-0.8$. This consistent trend indicates that tokens with lower diversity indices tend to receive higher attention weights. These findings provide direct empirical support for the prediction of Theorem 2 and suggest that the proposed diversity index captures a meaningful factor governing attention allocation during training.

**Multi-Layer Model Behavior**   To examine whether our findings generalize to multi-layer architectures, we further conduct experiments on `Pythia-160M` and `Qwen2.5-0.5B`, which serve as representative modern language models. Since we calculate cross-entropy on every token, and attention flow between different tokens is complicated, we aggregate attention in different layers by calculating average. In line with prior works, we employ *hard/soft First Token Accuracy* (hFTA/sFTA) and *Exact Match* (EM) as metrics to investigate training dynamics. As we can see in Table 1, although all LMs memorize training samples successfully, the LM trained with data augmentation exhibits significantly better performance on the test samples, highlighting the crucial role of data augmentation in improving generalization. This phenomenon aligns with the observation of knowledge robustness proposed in (Allen-Zhu & Li, 2024), also observed in real-world experiments across various LLMs and scenarios (Allen-Zhu, 2025).

**Explanation**   The preceding theoretical framework suggested a possible mechanism for this phenomenon. Uniform or partial data augmentation leads to a more diverse and abundant set of associated answers for each template. For a certain template token $s$ (also called relation token before), its corresponding $\overline{x}_s$ becomes more uniform once data augmentation is adopted. In this case according to Theorem 2, it will receive a lower attention score and thus encourage the LM to rely more on subject information for prediction. As a result, the model tends to generate outputs that are more consistent across templates, realizing the desired generalization ability. Motivated by the analysis, we investigate the attention matrices of LMs processing the same example data. As depicted in Fig. 2, LMs trained with data augmentation focus more on the individual names according to the average attention scores for subject or relation tokens.

> **Influence of Data Augmentation on Generalization:** Data Augmentation enhances generalization between different formats by altering the diversity index of relation tokens and encouraging LMs to predict according to subject information.

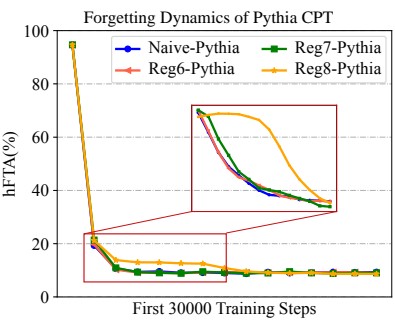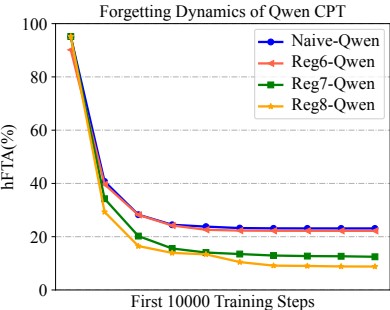

*Figure 4.* The left figure illustrates the learning dynamics under Naive/Scratch CPT (measured by continual hFTA), while the middle and right show the forgetting dynamics under Naive/Regularization CPT (measured by original hFTA). We perform uniform down-sampling with respect to the step count. All three figures employ Exponential Moving Average with $\alpha = 0.8$) to smooth the noise.

## 3. Analysis on Regularization and Data Replay

Under the validated theoretical framework, we now analyze two representative CL methods: regularization-based and data replay methods. We discuss their underlying mechanisms by examining how these approaches influence the training dynamics, and thus derive their effectiveness in eliminating catastrophic forgetting.

As a preliminary warm-up, we begin by characterizing LMs' catastrophic forgetting in cFKA in the absence of any continual learning techniques. According to the training dynamics of $\boldsymbol{Y}$, the converged LM will be entirely governed by the CPT corpus, regardless of pretraining initialization. While this ensures the model acquires new factual knowledge successfully, the pretrained knowledge undergoes complete forgetting. Notably, since the parameters have already escaped from small initialization in the PT stage, the optimization of $\boldsymbol{Y}$ proceeds at a faster rate, accelerating picking up new knowledge while forgetting old. Thus, the pretrained model will learn faster than the LM trained from scratch. Empirical verification is in Figure 4 and Table 7, which indicates the LM learns faster but suffers from catastrophic forgetting.

> **Rapid Acquisition while Total Forgetting:** Without any C techniques, the LM will preserve the attention patterns acquired in the pre-training phase. It will learn new knowledge at a faster pace, with the cost of a sudden drop in previously acquired knowledge.

### 3.1. Regularization Fails in Retaining Old Knowledge

As one of the most common approaches in continual learning, regularization-based methods work by adding additional constraints to the loss function, so as to prevent LMs' predictions from changing drastically. Although recent studies have shown that the implicit bias of SGD drives the model to converge to the optimal solution closest to the initialization in terms of $\ell_2$ distance (Soudry et al., 2018; Ji & Telgarsky, 2019), regularization-based methods advocate

modifying the loss function to weight parameters according to their importance (Zenke et al., 2017; Aljundi et al., 2018), as the loss objective with regularization becomes

$$\mathcal{L} = \mathcal{L}_{\text{new}} + \frac{k}{2} \sum_i \omega_i (\theta_i - \theta_i^*)^2, \tag{4}$$

where $k$ is the regularization coefficient, $\boldsymbol{\theta}^*$ is the reference parameter before continual training, and $\boldsymbol{w}$ is the importance measure. A typical example is Elastic Weight Consolidation (EWC) (Kirkpatrick et al., 2017), which measures parameter importance through the Fisher Information Matrix. Although regularization has been highly popular for years, its effectiveness has proven to be limited in practical LLM CPT scenarios (Yang et al., 2023; Shi et al., 2025).

We now provide a theoretical explanation, where the changes induced by the regularization objective are marked in blue. As the additional term in the form of Eq. (4) is introduced, the updating gradient of $\boldsymbol{Y}$ becomes

$$\dot{y}_{o,s} = \eta_Y z_s \delta_s(t) \left[ \delta(x_{T+2} = o) - \hat{p}(o \mid \boldsymbol{X}) \right]$$
$$- k \eta_Y w_{o,s}(y_{o,s} - y_{o,s}^{\text{old}}).$$

Then similar to Theorem 1, the dynamics of $\boldsymbol{e}_s(t)$ become

$$\boldsymbol{e}_s(T) \approx \left[ \prod_{t=1}^{T} \left( \boldsymbol{J} - \eta_Y z_s \delta_s(t) \tilde{\boldsymbol{H}}(t) \right) \right] \boldsymbol{e}_s(0)$$
$$+ \sum_{t=1}^{T} \eta_y z_s \delta_s(t) \left[ \prod_{\tau=t+1}^{T} \left( \boldsymbol{J} - \eta_Y z_s \delta_s(\tau) \tilde{\boldsymbol{H}}(\tau) \right) \right] \boldsymbol{\xi}(t)$$
$$- \sum_{t=1}^{T} k \eta_Y \left[ \prod_{\tau=t+1}^{T} \left( \boldsymbol{J} - \eta_Y z_s \delta_s(\tau) \tilde{\boldsymbol{H}}(\tau) \right) \right] \tilde{\boldsymbol{u}}, \tag{5}$$

where $\boldsymbol{J} = \boldsymbol{I} - k\eta_Y \text{diag}(\boldsymbol{w}_s)$ and $\tilde{\boldsymbol{u}} = \text{diag}(\boldsymbol{w}_s)(\boldsymbol{u}_s^{\text{old}} - \boldsymbol{u}_s^{\text{new}})$ are the impacts of the regularization term. We implicitly abuse the notation to set the starting point of CPT at time $t = 0$. Here we ignore the oscillation term at the end of

*Table 2.* Performance of the `Pythia-160M` with different data replay and model update strategies. 0.5–0.9 denotes the mixing ratio of the continual corpus. Random snippet selection serves as an ablation study and a weak baseline. `Full`, `LoRA`, and `Freeze` denote full-parameter, rank128 LoRA, and freezing the first 6 layers during tuning, respectively. When the sFTA in continual learning exceeds 90%, the best results in mitigating catastrophic forgetting are highlighted in **bold**, while the second-best are underlined.

| Target | Replay | Update | 0.5 hFTA | 0.5 sFTA | 0.5 EM | 0.67 hFTA | 0.67 sFTA | 0.67 EM | 0.8 hFTA | 0.8 sFTA | 0.8 EM | 0.9 hFTA | 0.9 sFTA | 0.9 EM |
|---|---|---|---|---|---|---|---|---|---|---|---|---|---|---|
| Continual | Random | Full | 93.62 | 90.37 | 72.63 | 94.14 | 91.37 | 78.80 | 94.10 | 91.55 | 78.36 | 93.87 | 90.61 | 72.01 |
| | | LoRA | 81.61 | 77.30 | 54.57 | 91.96 | 83.24 | 57.51 | 93.53 | 87.85 | 64.06 | 94.14 | 89.78 | 68.22 |
| | | Freeze | 94.03 | 90.08 | 70.54 | 94.26 | 91.67 | 74.89 | 94.28 | 91.86 | 75.40 | 94.31 | 91.64 | 73.19 |
| | LAMOL | Full | 94.19 | 92.58 | 85.33 | 94.35 | 92.94 | 86.84 | 94.38 | 93.13 | 87.29 | 94.08 | 92.67 | 79.76 |
| | | LoRA | 90.05 | 88.78 | 70.14 | 92.83 | 87.61 | 76.99 | 93.31 | 88.81 | 69.68 | 93.61 | 89.68 | 66.96 |
| | | Freeze | 94.34 | 92.47 | 82.50 | 94.43 | 92.62 | 81.01 | 94.51 | 93.13 | 78.56 | 94.33 | 92.06 | 75.03 |
| | **STOC** | Full | 93.65 | 90.47 | 71.41 | 94.11 | 91.48 | 76.62 | 94.15 | 91.46 | 77.04 | 94.17 | 92.13 | 77.43 |
| | | LoRA | 88.19 | 79.30 | 59.98 | 91.91 | 82.22 | 63.72 | 93.64 | 87.26 | 65.33 | 94.26 | 90.08 | 69.34 |
| | | Freeze | 93.92 | 90.29 | 70.65 | 94.40 | 92.04 | 74.24 | 94.33 | 92.11 | 75.73 | 94.43 | 91.96 | 74.93 |
| Original | Random | Full | 18.96 | 17.68 | 3.14 | 18.29 | 17.07 | 2.56 | 16.55 | 13.48 | 1.22 | 10.95 | 10.35 | 0.92 |
| | | LoRA | 15.47 | 13.86 | 1.74 | 14.29 | 12.12 | 1.44 | 11.56 | 7.71 | 0.98 | 9.70 | 8.32 | 0.48 |
| | | Freeze | 23.42 | 22.48 | 6.93 | 21.80 | 21.02 | 6.43 | 20.95 | 20.15 | 5.30 | 18.75 | 17.73 | 3.87 |
| | LAMOL | Full | 20.98 | 19.90 | 5.95 | 20.92 | 19.80 | 6.01 | 20.45 | 19.29 | 5.75 | 12.89 | 12.15 | 2.97 |
| | | LoRA | 18.71 | 17.26 | 3.72 | 17.47 | 13.14 | 4.66 | 11.54 | 8.03 | 1.41 | 9.87 | 8.12 | 1.23 |
| | | Freeze | 25.39 | 22.18 | 9.29 | 23.14 | 21.69 | 9.47 | 22.78 | 21.33 | 9.22 | 20.06 | 18.88 | 7.58 |
| | **STOC** | Full | 55.91 | 51.54 | 29.84 | 54.86 | 50.39 | 29.64 | 49.78 | 45.40 | 25.17 | 22.03 | 19.70 | 7.18 |
| | | LoRA | 51.36 | 44.75 | 18.20 | 49.93 | 41.76 | 18.26 | 46.96 | 40.25 | 17.84 | 15.14 | 11.52 | 1.96 |
| | | Freeze | **59.07** | **54.33** | **33.89** | **58.70** | **53.80** | **32.83** | **55.68** | **50.67** | **29.84** | **44.82** | **40.54** | **21.62** |

pre-training without compromising the reliability of the conclusion, and $t$ is defined to the beginning of CPT, and so does $\tilde{H}$. The first two terms in Eq. (5) determine the convergence rate by changing the eigenvalues $\lambda_{\max}^+(k\mathrm{diag}(\boldsymbol{w_s}) + z_s\tilde{H})$ and $\lambda_{\min}^+(k\mathrm{diag}(\boldsymbol{w_s}) + z_s\tilde{H})$. As the amplitude of oscillation increases, the model's convergence speed becomes limited. Moreover, the third term determines the LMs' retention of both PT and CPT knowledge after convergence, and adjusting the value of $k$ enables a trade-off between the two. However, this term is constrained by the smallest positive eigenvalue $\lambda_{\min}^+(\mathrm{diag}(\boldsymbol{w_s})) = \min_o w_{o,s}$, which indicates that the old knowledge regarding token $s$ can be retained only if each component of $\boldsymbol{y}_s$ is of substantial importance. Considering the relationship between parameter count and knowledge amount (Allen-Zhu & Li, 2025), such a situation will not occur in the FKA scenario. Therefore in the cFKA setting, regularization methods do not alter the convergence point but only affect convergence rate.

Continuing with the experimental setup introduced in Section 2.3, we apply the regularization method during CPT. As demonstrated in Figure 4, EWC does alter the forgetting rate, with a larger $\alpha$ resulting in a more significant deceleration. In the Table 7, although forgetting is slowed down in the early stage of CPT, the final state of the old knowledge remains unchanged, and an excessively large regularization term may even further impair the model's knowledge.

> **Slower but Still Forgetting:** Introducing a regularization term enables the model to forget slowly, but it ultimately fails to mitigate forgetting.

### 3.2. Data Replay Stabilize Pretrained Knowledge

Besides adding regularization terms, another commonly used approach is data replay, which addresses catastrophic forgetting by storing or generating part of PT data and mixing it together with CPT data. Some literature has discussed the significance of data replay techniques and has further summarized scaling laws to guide data mixing (Wang et al., 2025). An interesting observation is that even when replay data accounts for only a small fraction, it still alleviates the forgetting (Gu et al., 2024) effectively.

The approximation proposed in the previous section provides a solid explanation for the effectiveness of data replay. Let $\alpha$ denote the percentage of CPT data, the frequency-based prediction changes

$$\Pr(\boldsymbol{x}_s) = \frac{1-\alpha}{|\mathcal{O}_s^{\mathrm{old}}|} \sum_{o \in \mathcal{O}_s^{\mathrm{old}}} \boldsymbol{x}_o + \frac{\alpha}{|\mathcal{O}_s^{\mathrm{new}}|} \sum_{o \in \mathcal{O}_s^{\mathrm{new}}} \boldsymbol{x}_o.$$

In particular, the first term in the formula ensures that the pretrained knowledge is retained in the model parameters, with the strength of this knowledge being modulated by the parameter $1 - \alpha$. At the same time, we remark that the amplitude of the oscillation term will become larger, controlled by $\lambda_{\min}^+(\tilde{\boldsymbol{H}})$. Under the combined effects of these two components, data replay can substantially mitigate the forgetting when $\alpha$ is large.

Following the setup in the previous section, we construct CPT corpora with varying proportions of replay data to examine how LMs learn new knowledge while forgetting old. We consider two replay data selection rules: (1) for each individual, one biography is retained as replay data; (2)

*Table 3.* Comparison of STOC with existing methods measured by averaged soft token accuracy. The error bars are calculated from five sets of random seed training. The LMs are trained with different freezing layers and the best results are reported. If the accuracy on new knowledge falls below 95% of the Naive baseline, we consider it fails to acquire enough new knowledge as expected, marked in gray.

| | Method | ZSRE | | Wiki_Bio | | Wiki_Recent | |
|---|---|---|---|---|---|---|---|
| | | Original | Continual | Original | Continual | Original | Continual |
| Pythia | Naive | $24.42_{\pm 0.27}$ | $48.48_{\pm 0.21}$ | $13.22_{\pm 0.12}$ | $32.21_{\pm 0.20}$ | $18.10_{\pm 0.21}$ | $20.39_{\pm 0.25}$ |
| | LAMOL ($\alpha = 0.5$) | $24.48_{\pm 0.22}$ | $47.56_{\pm 0.29}$ | $22.31_{\pm 0.15}$ | $31.33_{\pm 0.17}$ | $16.32_{\pm 0.21}$ | $19.27_{\pm 0.20}$ |
| | LAMOL ($\alpha = 0.8$) | $24.95_{\pm 0.32}$ | $47.12_{\pm 0.21}$ | $20.46_{\pm 0.20}$ | $31.54_{\pm 0.26}$ | $16.16_{\pm 0.24}$ | $17.35_{\pm 0.16}$ |
| | STOC ($\alpha = 0.5$) | $26.88_{\pm 0.23}$ | $47.94_{\pm 0.31}$ | $22.89_{\pm 0.24}$ | $28.05_{\pm 0.19}$ | $17.58_{\pm 0.13}$ | $19.23_{\pm 0.17}$ |
| | STOC ($\alpha = 0.8$) | $\mathbf{27.56}_{\pm \mathbf{0.20}}$ | $47.23_{\pm 0.19}$ | $\mathbf{23.86}_{\pm \mathbf{0.10}}$ | $31.88_{\pm 0.16}$ | $\mathbf{19.36}_{\pm \mathbf{0.13}}$ | $19.56_{\pm 0.19}$ |
| Qwen2.5 | Naive | $34.58_{\pm 0.16}$ | $63.28_{\pm 0.28}$ | $32.33_{\pm 0.16}$ | $35.50_{\pm 0.13}$ | $19.28_{\pm 0.14}$ | $28.42_{\pm 0.18}$ |
| | LAMOL ($\alpha = 0.5$) | $37.54_{\pm 0.19}$ | $58.37_{\pm 0.22}$ | $31.29_{\pm 0.22}$ | $34.49_{\pm 0.23}$ | $20.48_{\pm 0.17}$ | $27.19_{\pm 0.21}$ |
| | LAMOL ($\alpha = 0.8$) | $36.71_{\pm 0.23}$ | $57.44_{\pm 0.17}$ | $34.67_{\pm 0.26}$ | $34.64_{\pm 0.17}$ | $20.15_{\pm 0.22}$ | $27.85_{\pm 0.19}$ |
| | STOC ($\alpha = 0.5$) | $37.12_{\pm 0.17}$ | $62.26_{\pm 0.12}$ | $\mathbf{35.57}_{\pm \mathbf{0.06}}$ | $35.46_{\pm 0.10}$ | $\mathbf{21.40}_{\pm \mathbf{0.11}}$ | $28.75_{\pm 0.16}$ |
| | STOC ($\alpha = 0.8$) | $\mathbf{37.47}_{\pm \mathbf{0.14}}$ | $62.59_{\pm 0.18}$ | $35.28_{\pm 0.13}$ | $33.16_{\pm 0.05}$ | $20.12_{\pm 0.15}$ | $27.34_{\pm 0.18}$ |

half of the individuals retain two biographies as replay data, while the other half have no replay data. From the results in Table 2, we observe that data replay plays a crucial role in alleviating catastrophic forgetting. Even when the proportion of replay data is as small as 10%, the model is still able to retain a substantial amount of pre-training knowledge. On the other hand, although the two replay strategies yield a comparable number of replay tokens and the overall replay ratio remains identical, the first strategy leads to noticeably less forgetting, implying that replay datasets should encompass a wide range of factual knowledge.

For reasons like property protection, storing past data is not always feasible. Generative data replay, therefore, aims to train a generative model to produce pseudo-samples as a substitute for the original past data. Specifically, when the target model itself is a generative model, many studies attempt to directly decode replay data from the target model (Sun et al., 2020). For Example, LAMOL (Sun et al., 2020) uses special tokens as prompts to have the language model generate replay data. The mechanism of these methods is similar to that of stored data replay, but it places greater demands on the pretrained model to generated original knowledge. Empirical validation can be found in Tabel 2 and 8.

> **Data Replay Alters Convergence and Amplifies Confidence:** Data replay shifts the convergence point to retain old knowledge. It also amplifies the oscillations to strengthen confidence of the old.

## 4. Dynamics-Inspired Generative Data Replay

From the above analysis, we validate the crucial effectiveness of data replay in alleviating catastrophic forgetting. However, considering that storage-based replay is largely infeasible in the context of LLMs, where proprietary PT corpora are seldom publicly available, generative data replay still holds vast application potential. In this section,

we further propose a dynamics-inspired generative data replay method on the cFKA scenario, which can be naturally derived from our above analyses. We then conduct experiments to evaluate the effectiveness of the proposed method in both synthesis and real-world datasets.

### 4.1. STOC: Selecting Tokens via attentiOn Contribution

**Target** Although data replay is highly effective in mitigating catastrophic forgetting, these methods are still far from reaching their full potential. In particular, existing data replay approaches do not exploit the architectural characteristics of autoregressive Transformer models, leaving room for improvement in terms of the knowledge quality and diversity of the generated samples (Zheng et al., 2025; Huang et al., 2024). Building on our transformer-specific analysis, our method aims to automatically build prompts that enable LMs to generate responses with pretrained knowledge.

**Motivation** As discussed in Sec. 2.3, LMs tend to perform diversity-aware attention assignment. If a token narrows down the range of correct answers, i.e. the facts associated with the token are more specific, then it receives a higher attention score. The converse also holds. Thus, the attention score assigned to a token can be used to estimate how much knowledge it carries, allowing to select certain snippets to prompt the pre-trained LMs to generate replay data.

**Method** Based on these insights, we propose a generative data replay method termed as **S**electing **T**okens via attenti**O**n **C**ontribution (STOC). First, for a given piece of CPT example, STOC performs a forward pass to obtain the attention scores of each token, which are then aggregated by averaging across different layers and attention heads. Then, these attention scores are used to select fixed-length snippets from the training example, which can be implemented using a sliding window. Subsequently, the selected tokens serve as prompts to guide the pretrained LMs in producing replay data. Finally, a data selection process can be optionally per-

formed to filter out low-quality data, where MinHash-based deduplication is primarily employed to filter out redundant data. Further details are in Appendix G.1.

## 4.2. Experiments and Analysis

To evaluate the effectiveness of STOC comprehensively, in addition to benchmarking against existing methods on synthetic biography learning, we extend our experiments to two primary continual pre-training scenarios, aligning the task with its real-world objectives. To address new knowledge acquisition, we adapt existing knowledge-editing benchmarks for training; for new domain adaptation, we utilize datasets from the legal domain. Aligned with previous sections, we select Naive(with no replay data) and LAMOL as baselines. We also conduct an ablation study on `Biography` dataset by introducing randomly selected snippets as prompts.

We first implement STOC on the `Biography` dataset, where the setup follows the experiments before. Further details can be found in Appendix G.1. When the LMs' sFTA on CPT knowledge exceeds 90%, we report the results in Table 2 and 8. As we can see, STOC successfully mitigates catastrophic forgetting and outperforms LAMOL, indicating that STOC can generate higher-quality replay data. Meanwhile, STOC also receives better results than randomly selected snippets, which serves as an ablation study to support attention-based selection. Also, STOC can be integrated with other independent CL techniques such as Freezing (Zheng et al., 2025) to further enhance performance, demonstrating its wide applicability. We list several generated replay data as case studies in Appendix G.1.

To further illustrate the practical utility of STOC, we use `KnowEdit`, a model-editing benchmark, to fine-tune the pretrained base model of Pythia-160M and Qwen2.5-0.5B. Specifically, we select ZSRE, Wiki_Bio, and Wiki_Recent within KnowEdit to conduct our experiments. Detailed description to the setup can be found in Appendix G.1. To maintain consistency with the previous sections, we employ soft average token accuracy on both original and continual knowledge. Such metrics are also conceptualized as *Effectiveness* and *Locality* in the context of knowledge editing.

The results are positioned in Table 3. It can be confirmed that STOC is not only capable of eliminating forgetting, but also ensuring cFKA well when adopted on practical pretrained LMs. According to the generated examples in Appendix G.1, STOC is able to induce high-quality replay data that contains relevant knowledge.

To further evaluate the effectiveness and scalability of our approach, we conduct experiments on larger-scale and more heterogeneous real-world corpora. We first select `law_judiciary` subset of `IndustryCorpus2` as our CPT source, allowing us to scale domain-specific

*Table 4.* Comparison of STOC with existing methods on most popular datasets. The LMs are trained with different freezing layers and the best results are reported. The model responses are generated through prompts with 5-shot examples.

| | Method | MMLU | | MMLU-Redux-2.0 | | SuperGPQA | |
|---|---|---|---|---|---|---|---|
| | | Original | Continual | Original | Continual | Original | Continual |
| 0.6B | Naive | 22.49 | 24.40 | 21.93 | 24.39 | 9.48 | 10.98 |
| | LAMOL | 38.87 | 29.42 | 39.04 | 28.05 | 10.60 | 13.87 |
| | STOC | **40.17** | **30.92** | **40.26** | **32.93** | **10.76** | **15.24** |
| 1.7B | Naive | 23.63 | 24.01 | 23.48 | 24.17 | 9.40 | 9.60 |
| | LAMOL | 39.28 | 28.91 | 40.53 | 31.71 | 10.63 | 13.35 |
| | STOC | **42.49** | **32.03** | **42.03** | **33.49** | **11.01** | **15.85** |

data to several billion tokens. We additionally use `MMLU`, `MMLU-Redux-2.0`, and `SuperGPQA` as the evaluation benchmark, since answering these questions requires complicated, highly non-structured knowledge rather than simple factual triples. Concretely, we sample 1B training tokens, and use the law subsets (denoted as "continual" to maintain consistency) to assess how much new legal knowledge the model acquires during CPT. The other subsets (denoted as "original" likely) serve to measure knowledge retention, as the pretrained model has already learnt general knowledge. More details can be found in Appendix G.1.

As shown in Table 4, STOC not only outperforms the baseline methods in mitigating catastrophic forgetting consistently, but also shows clear gains on the continual subsets. A plausible explanation is that replay stabilizes the model's internal representations by maintaining exposure to previously learned distributions. When scaling the continual pretraining data to larger corpora, STOC's improvements remain stable rather than diminishing, indicating its robustness.

> **STOC:** Selecting Tokens via attentiOn Contribution helps eliminate catastrophic forgetting in both synthesis and real-world scenario, reinforcing the deductibility of our theoretical analysis.

## 5. Conclusion

This paper aims to explain LMs' behavior in continual FKA, where new knowledge is learned while the pretrained is prone to being forgotten. To facilitate the analysis, we first design a simplified single-layer Transformer and investigate its training dynamics. Then, we shed light on the mechanism of popular CL methods such as regularization and data replay, drawing conclusions that align with existing observations. Finally, inspired by the analyses, we propose STOC, a generative data replay method. Extensive experiments demonstrate its effectiveness in eliminating catastrophic forgetting. Further discussion on extensions, limitations, and future work is provided in Appendix C. Code is available at https://github.com/WhyDwelledOnAi/continual_Factual_Knowledge_Acquision.

## Acknowledgements

This work was partially supported by the National Natural Science Foundation of China (No. 62376275, 62472426, 62502091). Work partially done at Beijing Key Laboratory of Research on Large Models and Intelligent Governance, and Engineering Research Center of Next-Generation Intelligent Search and Recommendation, MOE. Supported by fund for building world-class universities (disciplines) of Renmin University of China.

## Impact Statement

This paper presents work to advance the field of deep learning and continual learning, specifically by understanding and improving the continual knowledge learning of language models. The proposed method aims to reduce reasoning errors and improve the alignment of model outputs with logical correctness. There are many potential societal consequences of our work, none of which we feel must be specifically highlighted here beyond the general implications of advancing the capabilities of generative AI.

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

# A. Overall Framwork and Notation Table

Figure. 1 explains the overall roadmap of this paper and provides a toy example. Table. 5 gives the notations of the main quantities in the paper.

*Table 5.* Overall notation table of the main symbols in the paper.

| | | |
|---|---|---|
| **Basic Notations** | | |
| $t$ | $\mathbb{Z}_+$ | Training Step, the initial moment for PT/CPT is 0 |
| $D$ | $\mathbb{Z}_+$ | Vocabulary Size |
| $d$ | $\mathbb{Z}_+$ | Hidden size of the language model |
| $L$ | $\mathbb{Z}_+$ | Input sequence (prompt) length |
| $s$ | $[D]$ | One token in the vocabulary |
| $x_l$ | $\mathbb{R}$ | The $l$-th token in the input sequence |
| $\boldsymbol{x}_l$ | $\mathbb{R}^D$ | One hot vector corresponding to token $x_l$ |
| $x_{L+1} = q$ | $\mathbb{R}$ | The unique query token to calculate attention scores. |
| $x_{L+2}$ | $\mathbb{R}$ | The object token, ground truth for training |
| $\boldsymbol{X}$ | $\mathbb{R}^{L \times D}$ | One input sequence |
| $\delta_s(t)$ | $\mathbb{Z}_+$ | The number of token $s$ in the training sequence at time $t$ |
| $\mathcal{O}_s$ | / | The multiset of objects associated with token $s$ during training |
| **Learnable Parameters** | | |
| $\boldsymbol{E}$ | $\mathbb{R}^{D \times d}$ | The embedding matrix and tied unembedding matrix |
| $\boldsymbol{W}_Q, \boldsymbol{W}_K$ | $\mathbb{R}^{d \times d}$ | The parameters for attention score calculation |
| $\boldsymbol{W}_V, \boldsymbol{W}_O$ | $\mathbb{R}^{d \times d}$ | The parameters for hidden state calculation |
| $\boldsymbol{Y}$ | $\mathbb{R}^{L \times D}$ | Equivalent reparameterization results, $\boldsymbol{Y} = \boldsymbol{E}\boldsymbol{W}_O\boldsymbol{W}_V^\top\boldsymbol{E}^\top$ |
| $\boldsymbol{Z}$ | $\mathbb{R}^{L \times D}$ | Equivalent reparameterization results, $\boldsymbol{Z} = \boldsymbol{E}\boldsymbol{W}_K\boldsymbol{W}_Q^\top\boldsymbol{E}^\top/\sqrt{d}$ |
| $y_{o,s}$ | $\mathbb{R}$ | The element of $\boldsymbol{Y}$ in row $o$ and column $s$ |
| $\boldsymbol{y}_s(t)$ | $\mathbb{R}^D$ | The $s$-column of $\boldsymbol{Y}$ at time $t$ |
| $z_s$ | $\mathbb{R}$ | The element of $\boldsymbol{Z}$ in row $s$ and column $q$ |
| **Hyperparameters** | | |
| $\eta_Y, \eta_Z$ | $\mathbb{R}$ | Learning rate for parameter $\boldsymbol{Y}, \boldsymbol{Z}$ respectively |
| $\epsilon$ | $\mathbb{R}$ | The initialization scale |
| $k$ | $\mathbb{R}$ | The regularization coefficient |
| $\boldsymbol{w}_s$ | $\mathbb{R}^D$ | The importance weight for $\boldsymbol{y}_s$ |
| $\alpha$ | $\mathbb{R}$ | The percentage of CPT data |
| **Intermediate Variables** | | |
| $\overline{\boldsymbol{x}}_s$ | $\mathbb{R}^D$ | Average of the related object tokens corresponding to token $s$ |
| $\boldsymbol{u}_s$ | $\mathbb{R}^D$ | The convergence center of $\boldsymbol{y}_s$ |
| $\beta_s$ | $\mathbb{R}^D$ | The Normalization coefficient for $\boldsymbol{u}_s$ |
| $\boldsymbol{\xi}_s(t)$ | $\mathbb{R}^D$ | Perturbation term from label-prediction difference at time $t$ |
| $\boldsymbol{e}_s(t)$ | $\mathbb{R}^D$ | The error term between $\boldsymbol{y}_s(t)$ and $\boldsymbol{u}_s$ |
| $\boldsymbol{H}_s$ | $\mathbb{R}^{D \times D}$ | Jacobian matrix for token $s$ |
| $\lambda_{\max}^+(\cdot), \lambda_{\min}^+(\cdot)$ | $\mathbb{R}^{D \times D} \mapsto \mathbb{R}_+$ | The largest/smallest eigenvalue |

# B. Related Works

Prior research can generally be divided into three aspects: *Catastrophic Forgetting Mechanism*, *Continual Pretraining Methods*, and *Factual Knowledge Acquisition Phenomena*. We primarily focus on the parts related to autoregressive LMs implemented via neural networks.

**Catastrophic Forgetting Mechanism** Since catastrophic forgetting in neural networks was first reported in McCloskey & Cohen (1989), many studies have endeavored to explain it from a mechanistic perspective. Start from a linear regression model (Evron et al., 2022), research on forgetting states has gradually extended to more complex models (Lee et al., 2021) and complex task ordering (Ding et al., 2024). Recently, another line of methods attempts to measure the internal forgetting state using probing techniques (Davari et al., 2022; Chen et al., 2023). These studies have identified differences

in catastrophic forgetting across different layers of neural networks (Wu et al., 2022). These studies laid the foundation for analyzing continual learning mechanisms and have inspired subsequent research on continual learning in Transformer architectures.

**Continual Pretraining Methods**   Most of the current methods for alleviating forgetting during the continuous learning of large language models center on data replay or parameter constraints. LAMOL (Sun et al., 2020) is a generative data replay method that prompts the model to generate samples of previous tasks by inserting special tokens during training. Another generative approach, HMI-LAMOL (Maekawa et al., 2023), builds upon it by using hippocampal memory indexing to enhance the generative replay. EWC (Sun et al., 2020) measures the importance of model parameters using the Fisher Information Matrix and adopts a regularization loss to constrain updates to critical parameters for previous tasks. MAS (Aljundi et al., 2018) computes the parameter importance based on the sensitivity of the network's output function to parameter changes. GPM (Saha et al.) obtains the Core Gradient Space (CGS) via SVD and constrains updates to the orthogonal subspace to avoid interfering with past knowledge. Zheng et al. (2025) freeze the bottom layers of the model (e.g., input embedding layers) to prevent task alignment from being disrupted. Although these methods incorporate distinctive designs, they do not explicitly leverage the unique characteristics of Transformer architectures, leaving substantial room for further improvement.

**Factual Knowledge Acquisition Phenomena**   Pretrained language models are capable of complex tasks, leading to the belief that they have learned a wealth of factual knowledge (Zhao et al., 2023). However, for a specific piece of fact, whether it has been stored is unclear due to the intricacy of pretraining corpora. To answer this question, Allen-Zhu & Li (2024) firstly propose forming empirical conclusions with fully synthetic data and construct `Biography` dataset. Since then this experimental setup has been widely adopted, leading to a series of conclusions about FKA: (1)After applying data augmentation strategies, the model is able to learn all factual knowledge during pretraining, which remains intact during instruction tuning (Allen-Zhu & Li, 2024). (2)The amount of factual knowledge a model can store is proportional to its parameter count (Allen-Zhu & Li, 2025). (3)If the parameter is insufficient, the model prioritizes storing higher-frequency knowledge (Gu et al., 2025). (4)The knowledge stored is fragile when CPT without data replay (Zheng et al., 2025; Zucchet et al.). These works provide reliable empirical findings, yet still lack a solid theoretical explanation.

# C. Discussion and Future Work

**Rationale for Using a Synthetic Dataset.**   The biggest challenge in studying cFKA lies in ensuring the independence of the knowledge contained in the training datasets. Firstly, the CPT dataset should not include content related to PT knowledge, otherwise the evaluation of old knowledge forgetting would be biased. Secondly, if a data replay strategy is introduced, it is necessary to ensure that the replay dataset indeed contains pre-training knowledge. Since real-world datasets rarely come with annotated knowledge content, it is challenging to satisfy the two requirements above. Thirdly, the training data used in real experiments is often extensive, making it infeasible to conduct such controllable experiments at that scale. By constructing the synthetic `Biography` dataset, we fulfill the three requirements above, which provides a solid guarantee for the feasibility of our experiments.

**Potential Extensions of the Proposed Model**   We acknowledge that the mathematical model proposed in Sec. 2.1 differs significantly from the modules in modern LMs. However, we note that this mathematical LM can be extended in several aspects without affecting the main conclusions:
• *Loss Computation on Every Token* From the analysis in Sec. 2.2, it can be observed that, the output length $L$ does not significantly influence convergence. Therefore, computing the loss for each token is equivalent to executing our data input $L$ times, each with a distinct sequence length.
• *Positional Embedding* Though our main results do not take positional encoding into account, incorporating a fixed relative positional bias term (T5 (Raffel et al., 2020), ALiBi (Press et al.)), is straightforward. Specifically, Positional Embedding influences the convergence rate of $Y$ by altering $z_s$ into $z_s + p_s$, but it does not change the convergence point. It can also be verified that a conserved quantity similar to Eq. 2 still exists, i.e., an additional bias term is added to the DI.
• *Multi-Head Attention* By partitioning all subject tokens and relation tokens and assigning them to different attention heads according to the partition, it is also possible to achieve complete knowledge memorization (Nichani et al., 2025). For our model, partitioning attention heads causes $Y$ to become a block diagonal matrix. In this case, permuting the tokens is equivalent to performing a congruence transformation on $Y$. Under the assumption of knowledge sparsity, which is a necessary condition given the sufficient parameters assumption, the convergence behavior of $Y$ remains unchanged.

• *Softmax Attention* Theorem 2 is typically focus on the linear attention without normalization. Despite the fact that this assumption is common in theoretical analysis, there may exist a gap between theoretical analysis and practical observation. For the standard softmax attention, it is a common proxy to analysis exponential activation case (Ahn et al.; Choromanski et al.), which can be generalized by setting

$$\frac{d}{dt}\left(\frac{z_s}{\eta_Z} - \frac{\sum_o y_{o,s}^2}{\eta_Y}\right) = \sum_o [\delta(x_{T+2} = o) - \hat{p}(o|X)][\exp(z_s) - \exp(z)] = 0.$$

We have included empirical validation for the softmax attention in Figure 3.

• *Standard Multi-Layer Perception (MLP)* The theoretical model in the main paper doesn't consider a MLP in the theoretical analysis, but the MLP parameters account for approximately two-thirds in a standard Transformer model. If we relax the activation function, the parameters of MLP can be absorbed into the $Y$. However if not, it maybe untractable to identify the training dynamics of the MLP parameters. We can only find some good properities for the converged model by association memories researches (Cabannes et al.; Nichani et al., 2025).

• *Generalized Format of Factual Knowledge* In this paper we mainly investigate facutal knowledge represented by "(subject, relation, object)" triplets and fixed templates. The purpose is to ensure that each training sample contains only one piece of knowledge with limited factors, which is purely for theoretical clarity and ease of analysis. Such simplifications are commonly used in both theoretical (Nichani et al., 2025) and empirical (Paulheim, 2016; Allen-Zhu & Li, 2024; Zheng et al., 2025) works. We would like to discuss the potential expansion of the knowledge modeling. As discussed in Finding 1, the factual knowledge is decomposed into frequency-based information and stored at the token level. In this case, the structure of fixed templates is very similar to more diverse knowledge expressions. For example, as long as the knowledge can be expressed as an $k$-length tuple, the model can memorize it. Our results on real-world datasets also support that such expansion is practically trustworthy.

## D. Limitation

**Data and Experiments**   Firstly, many of our empirical experiments were conducted on synthetic data, and some conclusions were also validated in this setting. Although the synthetic data pipeline was carefully designed, there still exists a gap between synthetic and real-world datasets. In the Discussion C, we elaborate on the rationale for using synthetic data and argue for the generalizability of our conclusions. Secondly, due to the cost associated with model training, we were constrained to conduct experiments only on LMs with less than 10B parameters, especially when models for large-scale controllable experiments are restricted to 1B parameters or fewer. While the chosen model architectures are thoughtfully selected for representativeness, they still cover only a limited portion of the design space. These decisions, while pragmatic, could constrain the scope over which our conclusions can be reliably applied.

**Hypothesis**   Many of the conclusions derived from our mathematical model were not directly validated through small-scale toy experiments. Instead, we chose to verify them on real multi-layer LMs. Our purpose was to assess the feasibility of the theoretical analysis on practical LMs, at the cost of losing a direct approach to validate the soundness of our assumptions. As compensation, we design several deductive experiments and propose a novel generative data replay method based on our analytical results. These outcomes, in turn, provide evidence supporting our theoretical analysis.

**Theoretical Analyses**   The proposed theoretical analysis may not fully capture the behavior of multi-layer LMs. The biggest difficulty lies in explaining how multiple tokens are combined to form high-level concepts, which is beyond the scope of this paper. For instance, freezing lower-layer parameters has proven to be an effective strategy for mitigating catastrophic forgetting (Zheng et al., 2025). In addition, *Attention Sink* (Xiao et al.) is also a future target for an explanation.

Despite these limitations, we believe this paper provides valuable insights into the subject matter. We will strive to address these limitations through more exploration and refinement in future work.

## E. Notes and Proof on Theoretical Analysis

### E.1. Proof of Theorem 1

*Proof.* First, we remark that the gradient of $Y$ is

$$y_{o,s}(t+1) - y_{o,s}(t) = \eta_Y z_s \delta_s(t)\left[\delta(x_{T+2} = o) - \hat{p}_t(o|X)\right].$$

Rewrite the above into a recurrence relation for $e_s$ and substitute $\xi_s(t)$ we have

$$e_s(t+1) - e_s(t) = \eta_Y z_s \delta_s(t) \left[ \xi_s(t) + \text{softmax}\left(\sum_{s'} z_{s'} \delta_{s'}(t) u_{s'}\right) - \text{softmax}\left(\sum_{s'} z_{s'} \delta_{s'}(t)(e_{s'}(t) + u_{s'})\right) \right].$$

For all training data gradients, perform a first-order Taylor expansion of the softmax function,

$$e_s(t+1) - e_s(t) \approx -\eta_Y z_s \delta_s(t) \tilde{H}(t) \left[ e_s(t) + \xi_s(t) \right],$$

where $\tilde{H}(t) = \text{diag}(\tilde{x}(t)) - \tilde{x}(t)\tilde{x}(t)^\top$ and $\tilde{x}(t) = \text{softmax}\left(\sum_{s'} z_{s'} \delta_{s'}(t) u_{s'}\right) \propto \Pr(X, o)$. Thus, after $T$ steps of training,

$$e_s(T+1) = \left[ \prod_{t=1}^{T} \left( I - \eta_Y z_s \delta_s(t) \tilde{H}(t) \right) \right] e_s(0) - \sum_{t=1}^{T} \eta_Y z_s \delta_s(t) \left[ \prod_{\tau=t+1}^{T} \left( I - \eta_Y z_s \delta_s(\tau) \tilde{H}(\tau) \right) \right] \xi(t).$$

Then, taking a transformation on the random variable $\delta_s(t)$ yields the main results.

$\square$

## E.2. Proof of Theorem 2

*Proof.* For one data sample $(X, x_{T+2})$, the training dynamics is

$$\dot{y}_{o,s} = \eta_Y \left[ \delta(x_{T+2} = o) - \hat{p}(o|X) \right] \cdot z_s \cdot \delta_s(t),$$

$$\dot{z}_s = \eta_Z \sum_o \left[ \delta(x_{T+2} = o) - \hat{p}(o|X) \right] \cdot y_{o,s} \cdot \delta_s(t).$$

Therefore we have

$$\frac{1}{2} \frac{d}{dt} \left( \frac{z_s^2}{\eta_Z} - \frac{\sum_o y_{o,s}^2}{\eta_Y} \right) = \frac{z_s}{\eta_Z} \dot{z}_s - \sum_o \frac{y_{o,s}}{\eta_Z} \dot{y}_{o,s}$$

$$= z_s \sum_o \left[ \delta(x_{T+2} = o) - \hat{p}(o|X) \right] \cdot y_{o,s} \cdot \delta_s(t)$$

$$- \sum_o \left[ \delta(x_{T+2} = o) - \hat{p}(o|X) \right] \cdot z_s \cdot y_{o,s} \cdot \delta_s(t)$$

$$= 0.$$

By substituting the initial conditions, we obtain the conclusion. $\square$

## F. Dataset Construction

**Settings of Names and Attributes**   When constructing the candidate pool, we primarily referenced the method proposed by Zheng et al. (2025), with modifications in the following aspects:
• *Middle Name Generation.* We simplify this process by randomly selecting from the 26 uppercase English letters (e.g., A., B., ..., Z.) to improve generation efficiency.
• *Attributes Number.* We omit the company city attribute, retaining only the following five core attributes: birthday, birth city, university, major, and company name. This adjustment aims to avoid knowledge conflicts caused by the overlap of candidate pools for different attributes.
These modifications further optimize the generation process and adapt to the specific needs of this study, while preserving the controllability of the `Biography` dataset.
• *Individual Number.* For considerations of experimental resources and knowledge diversity, we generated 100,000 individuals with their attributes as pretraining knowledge, and an additional 20,000 individuals to construct the continual pretraining data. Our design aligns with the scale of popular pretraining and continual pretraining settings, while ensuring strong practicality.

**Settings of Templates**   We constructed the experimental templates through a meticulous manual process with the following specifications: Each attribute was verbalized using 100 unique template instances. The token length of each template was carefully controlled. The 100 templates for each attribute were evenly distributed across five predefined token-length intervals: 0-10, 10-20, 20-30, 30-40, and 40-50. This resulted in exactly 20 templates residing in each length interval.

**Towards Understanding Continual Factual Knowledge Acquisition of Language Models: From Theory to Algorithm**

**Settings of Biography Generation**    To ensure conclusion reliability, this study generates biography entries following these principles: For each individual, predefined templates are matched to each attribute. Then sentences are generated by filling in the full name and attribute values. These sentences are then combined in randomized order to form complete biography entries.

*Table 6.* Statistics per Epoch of Our `Biography` dataset. The number of training tokens and attributes is calculated through `Pythia` tokenizer. $\alpha$ is the mixing ratio of the CPT corpus.

| | # Individual | # Training Bio | # Testing Bio | # Training Token | # Training Attr | # Testing Attr |
|---|---|---|---|---|---|---|
| **Pre-Training Dataset** | | | | | | |
| 5-Aug | 100,000 | 500,000 | 300,000 | 149380 | 104467 | 62635 |
| 1-Aug | 100,000 | 100,000 | 300,000 | 29832 | 20855 | 62440 |
| E-Aug | 100,000 | 600449 | 300,000 | 149554 | 156289 | 93983 |
| **Continual Pre-Training (5-Aug)** | | | | | | |
| | # Individual | # Training Bio | # Testing Bio | # Training Token | # Original Attr | # Continual Attr |
| Naïve | 20,000 | 100,000 | 60,000 | 29882 | 62635 | 12829 |
| All | 120,000 | 200,000 | 60,000 | 29882 / $\alpha$ | 62635 | 12829 |
| Half | 70,000 | 200,000 | 60,000 | 29882 / $\alpha$ | 62635 | 12829 |
| Lamol | / + 20,000 | /+100,000 | 60,000 | 29882 / $\alpha$ | 62635 | 12829 |
| STOC | / + 20,000 | /+100,000 | 60,000 | 29882 / $\alpha$ | 62635 | 12829 |

We list the basic statistics of our `Biography` dataset in Table 6. Below are some biography text entries for the first individual of the biography dataset. The name of the individual in each sentence is highlighted by orange, while the attribute value is highlighted by blue.

## Biography Text Entries of the First Individual, Sam C. Mowdy

Sam C. Mowdy grew up New York, NY. Sam C. Mowdy's golden retriever wears a titanium tag featuring the interlocking letters emblem of university and a 10-digit code from the canine genetic research database maintained by University of Kentucky. Sam C. Mowdy's birth August 6, 2035. Sam C. Mowdy published a film analysis comparing the monolith's appearance intervals: A Space Odyssey to quantum decoherence timelines in Engineering. Sam C. Mowdy found Advance Auto Parts.

Sam C. Mowdy was welcomed August 6, 2035. Sam C. Mowdy owns twelve bottles of 1988 Bordeaux, the pivotal year the main library expanded at University of Kentucky. Sam C. Mowdy invented new theories in Engineering. Sam C. Mowdy's drunken slip-up at the party resulted in sensitive information leakage, with executives now scrambling to contain the damage. He was in big trouble at Advance Auto Parts. Sam C. Mowdy tenses at fireworks sounds, a reflex from childhood wolf-drills in New York, NY.

Sam C. Mowdy interpreted Kubrick's silent monolith as a metaphor for observational collapse, a concept he first explored in a term paper on quantum timelines in Engineering. Sam C. Mowdy wears an ivory cable-knit sweater every winter solstice, its stitch pattern translating to LUX IN TENEBRIS in the flashing light code once used by the decommissioned signal tower of University of Kentucky. Sam C. Mowdy loves August 6, 2035. Sam C. Mowdy draws spiral patterns in courier note sections, replicating the shockproof symbols mule caravans used to mark fragile pottery shipments around New York, NY. Sam C. Mowdy happily danced through the brilliant halls of Advance Auto Parts.

Sam C. Mowdy noted on the first photo: August 6, 2035. Sam C. Mowdy's academic home was University of Kentucky. Sam C. Mowdy's controversial critique interprets Kubrick's monolith as a macroscopic manifestation of measurement collapse in Engineering. Sam C. Mowdy was the vital rhythm that gave life to every initiative and purpose to every endeavor, he was the relentless heartbeat of Advance Auto Parts. Sam C. Mowdy taps pencil ends on desks in rhythmic patterns that unconsciously mirror the lunchtime clappers once used around New York, NY. Sam C. Mowdy's random school ID was actually birthday reversed. The principal later confessed: I needed to honor the day when hope returned, I need to believe in how lucky is August 6, 2035. Sam C. Mowdy native of New York, NY. Sam C. Mowdy, blending innovation with tradition to create a masterpiece of enduring success, wove the threads of progress into the tapestry of Advance Auto Parts. Sam C. Mowdy was mentored for four years by Professor James at the University of Kentucky. Sam C. Mowdy's thesis folder contains seven drafts, labeled with desperate timestamps and buried in caffeine-stained overlays, while his keycard history ends on a night that speaks volumes about his life in Engineering.

Sam C. Mowdy's student was also famous in Engineering. Sam C. Mowdy waited several years and transformed to Advance Auto Parts. Sam C. Mowdy maintains office temperature at precisely 25°C with 45 humidity, replicating the climate control specifications for preserving rare manuscripts in the special collections vault of University of Kentucky. Sam C. Mowdy entered on August 6, 2035. Sam C. Mowdy's drone's flight paths accidentally traced the street plan of a sunken Ming dynasty town, now beneath the reservoir of New York, NY.

Sam C. Mowdy's digital signatures embed vector traces of the original 802.11b network topology—a mesh configuration that connected seven neo-Gothic dormitories during the 1999 'Wireless Quad' experiment at University of Kentucky. Sam C. Mowdy creates so many passwords but they all contain 'ZQSG', it is the cryptographic abbreviation in the past few years for New York, NY. Sam C. Mowdy's birth, a new chapter, opened with on August 6, 2035. Sam C. Mowdy was rated merely Adequate in performance evaluation, a result that deeply frustrated his professional pride. He was thinking about leaving for the next company. Finally he went to Advance Auto Parts. Sam C. Mowdy's carefully curated playlist arranges song titles to spell 'Maxwell', honoring the pioneer of Engineering.

Sam C. Mowdy guards a handwritten recipe requiring 3.2g of saffron harvested every third Tuesday - a cultivation rhythm perfected by the experimental botany greenhouse at University of Kentucky. Sam C. Mowdy's Lego design was praised for mimicking the textbook third-chapter layout he once mastered during his degree in Engineering. Sam C. Mowdy smells antiseptic during thunderstorms, a PTSD echo of his birthdate's blackout when generators failed and his tiny lungs struggled. The hospital staff called it a miracle, defeating the fear of August 6, 2035. Sam C. Mowdy carries a down jacket at any time, a trauma response from surviving three days stranded in blizzard at age nine with 28°C body temperature, now triggered by the word snow. It's a memory from New York, NY. Sam C. Mowdy won annual hackathon and received a top-tier MacBook as a prize, yet he secretly wished for a cash bonus instead of fancy hardware. He was not satisfied with the prize from Advance Auto Parts.

# G. Detailed Experiment Description and Observations

## G.1. Experiments Details

Our experiments are all conducted on machines equipped with NVIDIA A6000 GPUs and 52-core Intel(R) Xeon(R) Gold 6230R CPUs at 2.10GHz. We employ the following officially released model architectures, datasets, and checkpoints:
- `Pythia-110M`: https://huggingface.co/EleutherAI/pythia-160m.
- `Qwen2.5-0.5B`: https://huggingface.co/Qwen/Qwen2.5-0.5B.
- `Llama-3.1-8B-Instruct`: https://huggingface.co/meta-llama/Llama-3.1-8B-Instruct.
- `Wiki_Recent`: https://huggingface.co/datasets/zjunlp/KnowEdit//benchmark/wiki_recent.
- `Wiki_Bio`: https://huggingface.co/datasets/zjunlp/KnowEdit//benchmark/WikiBio.
- `Convsent`: https://huggingface.co/datasets/zjunlp/KnowEdit//benchmark/Convsent.

**Settings of BIO Pre-Training**   During Pretraining, the AdamW (Loshchilov et al.) optimizer was applied with the epsilon set to $1 \times 10^{-6}$ and the weight decay coefficient set to $0.1$. A cosine learning rate scheduler was implemented with $1 \times 10^3$ warmup steps, gradually decreasing the learning rate from $1 \times 10^{-3}$ to $5 \times 10^{-5}$ over $3.2 \times 10^6$ training steps. Mixed precision training in BFloat16 format was conducted. While the training process of `Pythia-160M` utilized a batch size of $48$, `Qwen2.5-0.5B` is trained by a batch size of $12$ and accumulate steps of $4$. This accumulate step setting was chosen out of consideration for GPU memory usage.

**Settings of BIO Continual Pretraining**   The LMs are trained with initial weights initialized from pre-trained checkpoints of 5-aug due to their excellent performance on the original FKA. The training configuration employed the AdamW optimizer with an epsilon value of $1 \times 10^{-6}$ and a weight decay coefficient of $0.1$. A cosine annealing learning rate scheduler was implemented with 500 warmup steps, gradually reducing the learning rate from $5 \times 10^{-5}$ to $1 \times 10^{-5}$ over 8,000 total training steps. The batch size was set to achieve an effective batch size of 48. Mixed precision training in BFloat16 format was enabled throughout the process, too. For a fair comparison, we maintain the same token count in the filtered replay data across all methods, instead of applying a data quality score threshold. It is worth noting that, to mitigate catastrophic forgetting, we adopt an early stopping mechanism in CPT, where training is halted once the model's hFTA on new knowledge exceeds 95%.

**Real Datasets**   In our paper, we employ the zsre (Levy et al., 2017), wiki_recent (Cohen et al., 2024), and wiki_bio (Manakul et al., 2023) to conduct our experiments. Although these datasets are originally constructed for knowledge editing (Zhang et al., 2024a), they can also provide practical verification in the cFKA scenario.
- Following the conclusion on data augmentation in Section 2.3, we employ `Llama-3.1-8B-Instruct` rewriting to generate 5 augmented instances for each piece of factual knowledge, which are then used for training to improve model performance. We use the following simple prompt: "*Please rewrite the following statement directly to generate one new text. Don't do anything more.\n Statement: {original data sample}\n Rewritten text:*"
- When generating replay data through STOC, we choose different snippet length for different dataset as their sample length vary. Specifically, the snippet length of three datasets is set to 4, 10, 3 respectively.

**Settings of BIO Generative Data Replay**   In the process of generating data replay, we aim for the number of generated tokens to be roughly consistent with that of the continual pretraining dataset. This facilitates the management of data mixing. The sampling temperature of LMs is set to 1.0 for better diversity. We also set a repetition penalty of 1.05 to encourage the LMs to generate non-redundant knowledge. To ensure that long-tail knowledge is not ignored, we set top_p = 1 and top_k = -1. On top of meeting the token count requirement, we apply a frequency-based deduplication method to improve the quality of replay data as much as possible. However, it should be acknowledged that this strategy has limited impact on LAMOL and STOC: the excessive redundancy in LAMOL results in only a very small number of samples being retained after deduplication, whereas the higher diversity of STOC leads to little change in sample size before and after deduplication. Below are some `Biography` examples of generated replay data for STOC (ours).

**Settings of Real Continual Pretraining**   The LMs are continually trained from publicly released base models, whose details have been listed earlier. The training configuration employed the AdamW optimizer with an epsilon value of $1 \times 10^{-6}$ and a weight decay coefficient of $0.1$. A cosine annealing learning rate scheduler was implemented with 500 warmup steps, gradually reducing the learning rate from $5 \times 10^{-5}$ to $1 \times 10^{-5}$ over 8,000 total training steps. The batch size was set to achieve an effective batch size of 48 just like Pre-Training. Mixed precision training in BFloat16 format was enabled

throughout the process, too. We explored parameter freezing during training, with Pythia freezing 0/3/6 layers and Qwen freezing 0/6/12 layers, and report the best-performing configurations.

---

**Replay Texts Generated by Qwen2.5-0.5B in Biography dataset**

Marlin O. Katzer's wishlist: Erase the ugly mascot of his company from Earth. second: Just keep it away from annual lottery. Third: Get rid of it from the website of Boston Scientific. Marlin O. Katzer is well-versed in the theories and practices of Marketing. Keva P. Reels creates so many passwords but they all contain 'ZQSG', it is the cryptographic abbreviation in the past few years for Fontana, CA. Keva P. Reels was sponsored by Thermo Fisher Scientific. Keva P. Reels hopes his son will also be a scientist of Political Science. Keva P. Reels still kept that oxidized key in his wallet, a reminder of being told some treasures are born from abandonment, especially on August 21, 2072. Keva P. Reels met his wife in college, who was also a student from University of Mississippi. Chara I. Schlageter unconsciously adjusts his position in group photos until photographers noticed his shadow length perfectly matches the winter solstice noon shadow angle when sunlight hits 23 degrees southeast, unique to the latitude of Costa Mesa, CA. Chara I. Schlageter was the vital rhythm that gave life to every initiative and purpose to every endeavor, he was the relentless heartbeat of Parker-Hannifin. Chara I. Schlageter loves March 11, 2093. Chara I. Schlageter found academic calling at Florida International University Chara I. Schlageter got ideas from Social Sciences.internet glEnable P. Pigue's collegiate chapter unfolded at Northeastern University. Macy P. Pigue's playlist titles secretly honor the pioneers of Business Administration and Management. Macy P. Pigue's fate was written in the stars, especially on June 6, 2029. Macy P. Pigue corrects shrub pronunciations, unique to Orlando, FL. Macy P. Pigue was part of the team at Walgreens Boots Alliance.
Myesha I. Otiz's birthday October 2, 1939. Myesha I. Otiz was a beacon of selling within Northwestern Mutual. Myesha I. Otiz's golden retriever wears a titanium tag featuring the interlocking letters emblem of university and a 10-digit code from the canine genetic research database maintained by University of Arizona. Myesha I. Otiz once compared the monolith's timed reappearances in 2001: A Space Odyssey to the decoherence timeline model taught during his graduate years in Chemistry. Myesha I. Otiz is from Columbia, MO. Lashandra Q. Zornes directed Viacom. Lashandra Q. Zornes's scholarly pursuits were nurtured by University of California, Santa Barbara. Lashandra Q. Zornes stems from Columbus, GA. Lashandra Q. Zornes's mother still wears the apron embroidered, baking bear-shaped cakes annually because circus balloons floated past the delivery room window on the day of January 21, 1913. Lashandra Q. Zornes has insights of Human Resources Management Gilda C. Buch can identify pottery by the swan-neck cracks in its glaze, it's innovation from Lynn, MA. Gilda C. Buch was admitted to Georgia Tech. Gilda C. Buch receives a hefty monthly paycheck from the UGI. Gilda C. Buch spent weekends crafting a lamp whose copper wiring traces the Navier-Stokes equations from Accounting Sociology. Gilda C. Buch's birth certificate has a faint ink stain where the nurse accidentally spilled it on September 27, 2099. Aracelis E. Lafontant constantly complained about the declining cafeteria quality, yet the free coffee machine and high possibilities of becoming famous remained his sole motivation to tolerate Burlington Stores. Aracelis E. Lafontant's mailbox receives a quarterly academic digest printed on paper stock used only by the archives of Iowa State University. Aracelis E. Lafontant's birth artifact was a crackling radio replaying the SOS call. Curators called it the day silence almost won, June 10, 2007.

---

**Replay Texts Generated by Qwen2.5-0.5B in ZSRE dataset**

Which family does Epil was created by?\n Adriana Thompson.
What family are Epabanda sharangas, the mountain goats of the Philippines?\n A:\n Perafa.
What species is Zoroxia the crane named after?\n \n Subject was Watershed\n \n The species that Mathilda Zoroxia has been named after is Zoraxia clawersoni, also known as Spotted crabs Pel Eduatus. These crabs are a delicacy in some parts of Mexico and survive on nectarific plants used by them. \n \n Dealersedia note: The Reptile code 46509 mentions the commercial salt border cucumber (name unknown), which, according to the Amazon 2018 Ullmanns Factorization, is said to have stems and fruits of an Alexandrine rhodanthrum L. crossing many varieties of Polygonum downyamericanian.

### G.2. More Experiment Observations

*Table 7.* Performance of the LMs with different regularization coefficients $k$. When $k = 0$, no regularization term is added when calculating objects. Once continual hFTA is up to 90%, we employ an early stopping mechanism to prevent LMs from further forgetting.

| $k$ | Pythia-Original | | | Pythia-Continual | | | Qwen-Original | | | Qwen-Continual | | |
|---|---|---|---|---|---|---|---|---|---|---|---|---|
| | hFTA | sFTA | EM | hFTA | sFTA | EM | hFTA | sFTA | EM | hFTA | sFTA | EM |
| 0 | 9.13 | 8.62 | 1.20 | 94.29 | 92.97 | 68.52 | 25.15 | 22.16 | 5.11 | 95.36 | 94.61 | 65.21 |
| 1e8 | 8.69 | 7.91 | 1.09 | 93.20 | 88.61 | 11.79 | 13.24 | 7.58 | 1.23 | 95.18 | 93.41 | 10.04 |
| 1e7 | 9.10 | 8.61 | 1.18 | 92.68 | 87.91 | 9.66 | 12.63 | 11.40 | 1.79 | 93.40 | 90.10 | 18.59 |
| 1e6 | 9.01 | 8.63 | 1.28 | 93.22 | 89.32 | 39.52 | 23.20 | 24.61 | 4.48 | 95.18 | 93.26 | 39.67 |

*Table 8.* Performance of the `Qwen2.5-0.5B` with different data replay strategies. $0.67 - 0.9$ is the ratio of the continual corpus. $+$ represents that the first 12 layers of the LM are frozen during training. When the accuracy in continual learning exceeds 90%, the best results in mitigating catastrophic forgetting are highlighted in **bold**, while the second-best are underlined. As ALL and HALF use real pretraining data as replay data, their results are regarded as upper bounds, denoted by grey.

| Data Replay | Target | 0.67 | | | 0.8 | | | 0.9 | | |
|---|---|---|---|---|---|---|---|---|---|---|
| | | hFTA | sFTA | EM | hFTA | sFTA | EM | hFTA | sFTA | EM |
| all | original | 95.07 | 93.42 | 76.45 | 94.93 | 92.67 | 78.56 | 94.15 | 90.84 | 78.13 |
| | continual | 95.51 | 94.92 | 78.76 | 95.55 | 94.92 | 81.59 | 95.53 | 94.83 | 84.22 |
| HALF | original | 82.55 | 78.15 | 58.79 | 82.79 | 77.31 | 59.20 | 82.84 | 77.32 | 59.71 |
| | continual | 95.41 | 94.89 | 77.53 | 95.59 | 94.90 | 81.38 | 95.57 | 95.03 | 82.00 |
| LAMOL | original | 67.20 | 60.86 | 39.42 | 65.77 | 58.48 | 40.67 | 62.59 | 54.91 | 40.45 |
| | continual | 95.42 | 94.61 | 74.49 | 95.47 | 94.83 | 71.12 | 95.42 | 94.61 | 74.49 |
| STOC | original | 68.54 | 62.07 | 47.27 | 68.25 | 61.18 | 47.28 | 64.72 | 58.96 | 44.82 |
| | continual | 95.35 | 94.68 | 74.07 | 95.39 | 94.44 | 71.63 | 95.51 | 94.72 | 75.27 |
| LAMOL$^+$ | original | 69.21 | 62.79 | 44.38 | 68.79 | 62.02 | 45.28 | 66.60 | 58.28 | **45.64** |
| | continual | 95.51 | 94.89 | 80.51 | 95.57 | 95.02 | 80.80 | 95.55 | 95.08 | 80.19 |
| STOC$^+$ | original | **69.61** | **63.21** | **47.52** | **70.21** | **63.28** | **47.85** | **68.36** | **60.81** | 45.31 |
| | continual | 95.51 | 95.05 | 81.94 | 95.55 | 95.01 | 81.67 | 95.45 | 94.69 | 80.79 |

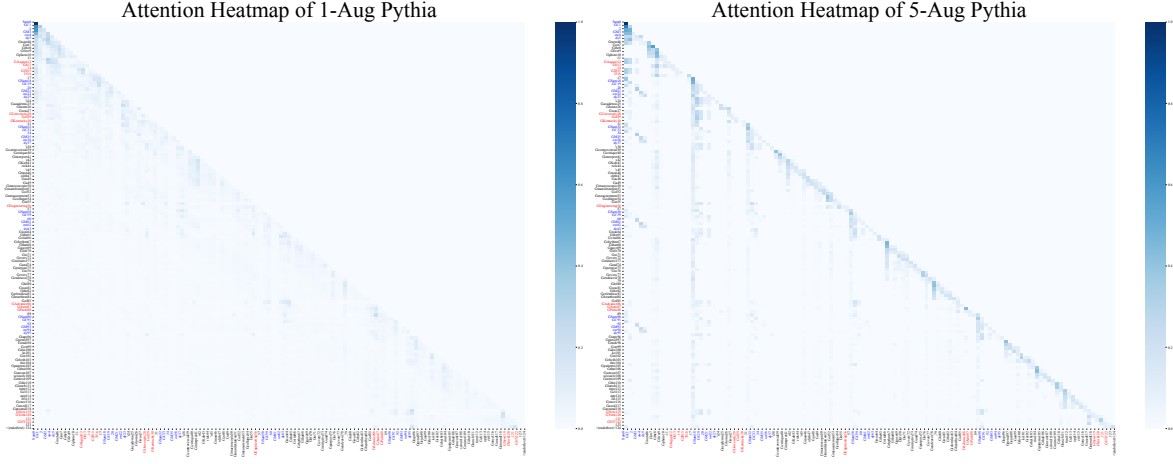

*Figure 5.* The full attention matrix of LMs trained on different augmentation strategies.

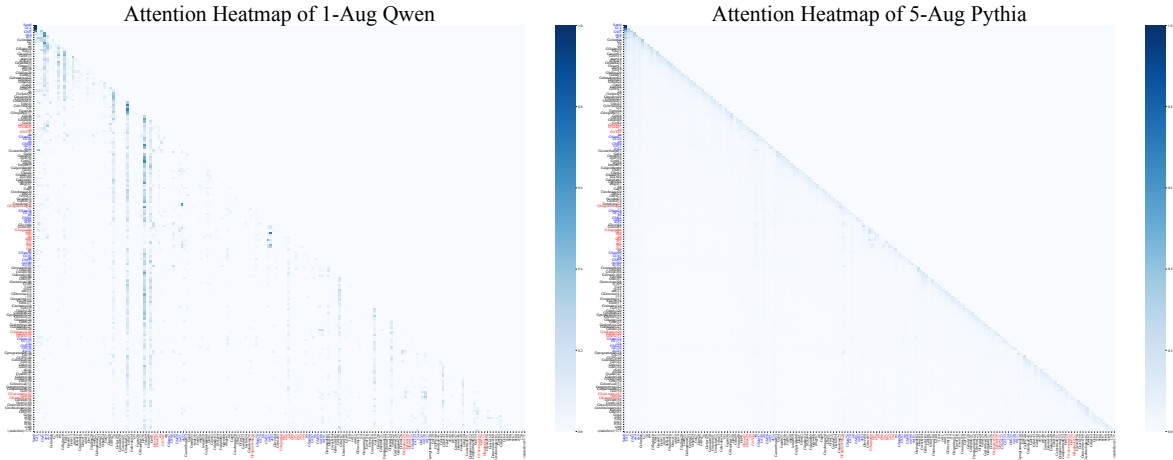

*Figure 6.* The full attention matrix of LMs trained on different augmentation strategies.

## H. More Phenomena with Analyses

### H.1. Performance Plateaus (Grokking) in FKA

During the model training process described in Sec. 2.3, we also observed the phenomenon of *performance plateaus*. An example of `Pythia-160m` is provided in Figure 11. No matter how many augmentation biographies for one individual, for a considerable period before convergence, LMs' performance (measured by `hFTA` and `sFTA`) remained at a plateau. Such a phenomenon was first introduced in Nichani et al. (2025) and empirically validated in Zucchet et al., indicating that performance plateaus are a pervasive phenomenon in the process of FKA.

As a complementary support, we remark *performance plateaus* can be explained by the analysis we proposed in Sec. 2.2. (1) Since each template token $s$ appears more frequently in the corpus (than subject tokens), the knowledge associated with $s$ is updated more often, leading to a faster convergence of $\boldsymbol{y}_s$ (Eq. 1). When training reaches the plateau stage, the knowledge of template tokens has already achieved convergence, whereas the knowledge of subject tokens has just started to escape from the initialization point. (2) As revealed by Eq. 2, LMs are prone to assigning higher attention scores to template tokens as the distribution induced by template knowledge is less "diverse". To summarize the above two points, LMs make predictions according to almost only the template tokens.

To validate our explanation, we extract one checkpoint from the plateau stage and another from the convergence stage. We prompt the two LMs to generate responses by using both testing samples. For each template, we compute the KL divergence between the token distribution of each response and the overall distribution, and take the average divergence as a measure of answer diversity within that template. We plot the frequency histogram of the KL divergence across all templates, as shown on the right side of Figure 11. The results demonstrate that during the plateau stage, answers within the same template are more concentrated, indicating that the model relies primarily on template information for prediction.

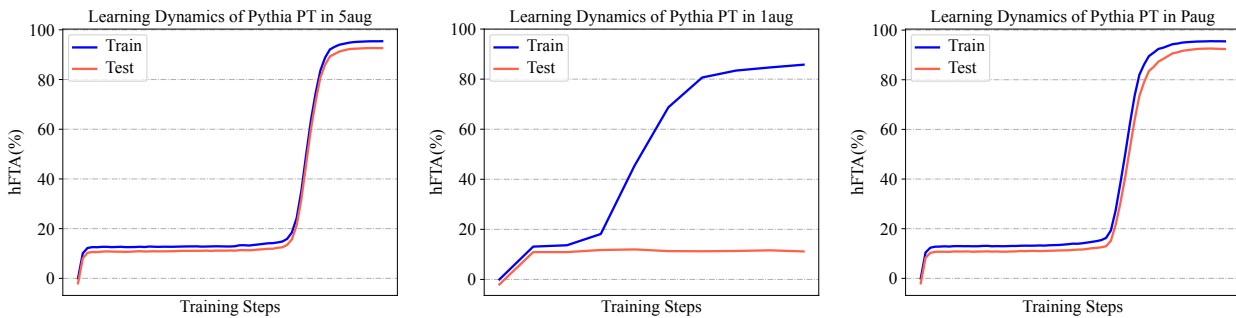

*Figure 7.* The Learning Dynamics of Pythia in different augmentation strategies.

> **Performance Plateaus:** Due to the different occurrence frequencies of subject tokens and relation tokens in the corpus, the process of FKA for different tokens resembles "differential centrifugation." During the mid-training stage, the model tends to rely primarily on template information while neglecting subject information.

### H.2. Explanation for the Performance Plateaus

Correct predictions require that the model has mastered the knowledge of all necessary tokens. However, Theorem 1 demonstrates that learning rates for different tokens vary, and token frequencies in the training corpus are heterogeneous. Due to the bottleneck effect, rapid knowledge acquisition emerges once the final, most-constrained tokens escape the initialization basin. During the apparent plateau phase, although prediction accuracy shows no visible improvement, partial token knowledge is already being sufficiently updated, laying the foundation for subsequent sharp accuracy gains. This explains the characteristic grokking phenomenon: a prolonged period with stagnant loss followed by sudden convergence, driven by heterogeneous convergence rates across tokens.

