# OpenReview forum: "Towards Understanding Continual Factual Knowledge Acquisition of Language Models: From Theory to Algorithm"
_ICML.cc/2026/Conference — ICML 2026 regular_

### Official Review · Reviewer_snZb · 2026-03-11

**Soundness:** 3
**Presentation:** 3
**Significance:** 3
**Originality:** 3
**Overall Recommendation:** 4
**Confidence:** 3

**Summary:**

The authors discuss the concept of continual Factual Knowledge Acquisition (cFKA) in language models and propose a theoretical framework based on a single-layer Transformer to characterize training dynamics under CPT. The authors explain why regularization-based methods fail to prevent catastrophic forgetting while data replay succeeds, through formal analysis of parameter convergence behavior. Building on these insights, the paper proposes STOC, a generative data replay method that selects high-attention token snippets as prompts to elicit pretrained factual knowledge.

**Compliance With Llm Reviewing Policy:**

Affirmed.

**Final Justification:**

Most of my concerns have been addressed, and I have accordingly increased my score.

**Key Questions For Authors:**

- It would strengthen the theoretical contribution to include at least one small-scale toy experiment (e.g., a 2-layer Transformer with softmax attention) that directly tests whether the conserved-quantity relationship in Theorem 2 or the convergence rate prediction of Theorem 1 holds approximately, rather than relying solely on deductive validation through downstream tasks.

- An ablation replacing STOC's attention-based snippet selection with random or frequency-based selection would help isolate how much of the observed benefit comes from the theoretically motivated selection criterion versus the general benefit of using CPT-derived prompts for generative replay.

- The inconsistent results in Tables 2 and 3 warrant discussion. Identifying the conditions under which STOC's advantage is robust versus marginal would make the practical guidance of the paper more actionable.

- There is a dangling sentence fragment at the start of Section 2 ("that our theoretical analysis can provide..."), and "architechure" (Appendix C) should be "architecture."

**Limitations:**

yes

**Strengths And Weaknesses:**

**Summary of Strengths:**

- Theoretically grounded explanation of an empirically observed gap. The distinction between regularization merely modifying convergence rate versus data replay shifting the convergence point is a clean and actionable insight, supported by both formal derivation and controlled experiments.

- Algorithm naturally derived from theory. STOC's use of attention scores as proxies for token-level factual specificity follows directly from Theorem 2's Diversity Index formulation, giving the method a principled motivation that distinguishes it from heuristic replay approaches.

- Diverse and reasonably scaled evaluation. Experiments span synthetic biography data, knowledge-editing benchmarks, and a large-scale domain adaptation setting, covering the main CPT scenarios with two representative model families.

**Summary of Weaknesses:**

- The theoretical framework relies on assumptions with limited empirical validation. The core analysis assumes ηY ≫ ηZ, linear attention without normalization, and tied embedding/unembedding matrices. These are acknowledged simplifications, but the paper's extension from single-layer theory to multi-layer LMs proceeds primarily via deductive experiments rather than direct verification. It is unclear whether the conserved quantity in Theorem 2 or the convergence characterization of Theorem 1 survives even qualitatively under standard softmax attention and untied weights. The Appendix discusses these extensions at a high level, but the gap between the theoretical model and the empirical setting remains underspecified.

- The mechanism connecting STOC's snippet selection to forgetting mitigation is not formally closed. STOC is motivated by the claim that high-attention tokens carry more factual specificity (per Theorem 2/DI). However, this relationship is established for the simplified single-layer model under linear attention, while STOC is applied to multi-layer models with aggregated cross-layer attention scores. The paper does not verify that attention scores in modern LMs reliably predict the factual specificity of snippets, nor does it ablate whether simpler selection heuristics would perform comparably.

- STOC's advantage over LAMOL is modest and inconsistent across settings. In Table 2, STOC without freezing underperforms LAMOL on original knowledge retention at mixing ratio 0.5 across both Pythia and Qwen, and its margin over LAMOL narrows or reverses in several cells of Table 3. The paper does not discuss these cases, leaving it unclear under what conditions STOC's attention-based selection reliably adds value over the simpler task-token prompting of LAMOL.

---

> ### Author Rebuttal · Authors · 2026-03-30
>
> Thanks for acknowledging our work and the patient, constructive comments. Please kindly find point-to-point responses below.
>
> > **W1, W2, Q1** The theoretical framework relies on assumptions with limited empirical validation. The core assumptions are acknowledged simplifications, but the paper's extension from single-layer theory to multi-layer LMs proceeds primarily via deductive experiments rather than direct verification. It is unclear whether the conserved quantity in Theorem 2 or the convergence characterization of Theorem 1 survives even qualitatively under standard softmax attention and untied weights. Thus, the mechanism connecting STOC's snippet selection to forgetting mitigation is not formally closed.
>
> Thanks for the question. We acknowledge that our theoretical analysis relies on several assumptions. As you have noted, these are widely recognized simplifications in the literature. Based on these assumptions, we derive meaningful theoretical conclusions that provide valuable insights. We have discussed the potential effects of some components in modern Transformers in Appendix C.  Therefore, we maintain that this work contains a valid and substantive theoretical contribution.
>
> On Multi-Layer Direct Verification: Analyzing parameter dynamics in multi-layer settings remains a fundamentally challenging task. Such direct verification on multi-layer models is infeasible cause we cannot analytically solve for the convergence points. This explains our choice to employ deductive experiments rather than parameter-level direct verification. Though the micro-level conclusions about parameter dynamics may not directly verified, the results in Sec. 2.3 show that the macro-level model behavior is substantially aligned. Thus, our theoretical framework still gets empirical support from practical multi-layer language models.
>
> To further address your concerns, we have retrained a model following the experimental setting in Section 2.2 (Last Token Prediction) and computed the DI according to Equation 3. This provides a direct validation of our theoretical predictions at the token level. For detailed results and discussion, please refer to our response to Reviewer YhDt (Q3 Part).
>
> > **W3, Q3**  STOC's advantage over LAMOL is modest and inconsistent across settings. In Table 2, STOC without freezing underperforms LAMOL on original knowledge retention at mixing ratio 0.5 across both Pythia and Qwen, and its margin over LAMOL narrows or reverses in several cells of Table 3.
>
> Thanks for the question about the performance, yet we respectfully disagree with the characterization that STOC's advantage is modest and inconsistent.
>
> In Tab. 2: At mixing ratio 0.5 without freezing, STOC's results (55.91/51.65/29.84) are significantly higher than LAMOL's (20.98/19.9/5.95). This contradicts the claim of underperformance.
> In Tab. 3: We provide error bars derived from five repeated runs, which demonstrate that STOC outperforms LAMOL with statistical robustness. Also, we remark that soft average token accuracy is a less-sensitive metric, which is also a reason that the advantage of STOC seems marginal to LAMOL.
>
> Based on these results, we maintain that STOC demonstrates clear and substantial superiority over LAMOL. The performance improvements are neither modest nor inconsistent; rather, they reflect the genuine benefits of our proposed method.
>
> > **Q2** An ablation replacing STOC's attention-based snippet selection with random or frequency-based selection would help isolate how much of the observed benefit comes from the theoretically motivated selection criterion versus the general benefit of using CPT-derived prompts for generative replay.
>
> Due to space constraints in this rebuttal, we refer the reviewer to our detailed response to Reviewer TKU1 (Q2 Part), where we provide validation of random snippet selection.  Regarding frequency-based selection, we opted not to pursue this approach for the following reason: On synthetic datasets, frequency-based N-gram statistics would result in template fragments dominating the selection, as they occur most frequently. Simultaneously, the frequencies of various name combinations are uniformly distributed, making them indistinguishable from a frequency perspective. Consequently, we conclude that frequency-based selection would not constitute a robust snippet selection strategy for this task.
>
> > **Q4** There is a dangling sentence fragment at the start of Section 2 ("that our theoretical analysis can provide..."), and "architechure" (Appendix C) should be "architecture.
>
> Thank you for pointing out these errors. The dangling sentence fragment should read: "Results indicate that our theoretical analysis can provide..." We will promptly correct these typographical errors.
>
> ---
> Thanks again for the valuable comments. Hope our responses address your concerns. Please feel free to raise any further questions.

---

> > ### Author Rebuttal · Reviewer_snZb · 2026-04-04
> >
> > Thank you for the detailed rebuttal. While most concerns have been addressed, two issues remain.
> >
> > Regarding W1/W2/Q1, the authors' reliance on macro-level behavioral alignment as a substitute for direct theoretical verification is understandable given the difficulty of analyzing multi-layer models, but it nonetheless leaves the core assumptions of Theorem 2 — particularly the Diversity Index formulation under standard softmax attention — without independent validation.
> >
> > Regarding W3/Q3, while the authors cite substantial performance gains over LAMOL at mixing ratio 0.5, the abnormally low retention scores of LAMOL (~20%) in that setting are left unexplained, making it difficult to fully attribute the observed gap to the proposed method's superiority rather than potential implementation or tuning issues on the baseline side.

---

> > > ### Author Response · Authors · 2026-04-05
> > >
> > > Thank you for your detailed feedback. We would like to provide further clarification on the following points.
> > >
> > > **Regarding Generalization to Softmax Attention:** Following your suggestions, we have conducted additional validation experiments using standard softmax attention. All experimental settings remain consistent with those described in the paper, except for the attention mechanism. The resulting scatter plots are available at: \url{https://anonymous.4open.science/r/continual_Factual_Knowledge_Acquision-63B1/asset/di2attn_softmax.pdf}. The results demonstrate that even with standard softmax attention, the parameter Z and token-level Diversity Index maintain a significant negative correlation (Pearson and Spearman coefficient approximately -0.8). These findings support the generalizability of our theoretical analysis across different attention mechanisms.
> > >
> > > **Regarding the LAMOL Performance:** The LAMOL implementation had been originally provided in Appendix G.1 of the paper. We welcome any meaningful suggestions on the baseline implementation. As also discussed with Reviewer TKU1,
> > > we attribute LAMOL's suboptimal performance to the limited diversity and specificity of the prompts used for generating replay data. The constrained prompt design results in insufficient diversity of generated samples and fails to leverage the architectural properties of Transformers. Although our evaluation includes a deduplication step, LAMOL's effectiveness remains fundamentally poor by the limited prompt diversity employed in the baseline implementation.
> > >
> > > We hope these clarifications adequately address your concerns and look forward to your further feedback.

---

### Official Review · Reviewer_bpZT · 2026-03-12

**Soundness:** 3
**Presentation:** 2
**Significance:** 3
**Originality:** 3
**Overall Recommendation:** 4
**Confidence:** 3

**Summary:**

The paper is trying to understand how the replay method in continuous factual knowledge learning prevents catastrophic forgetting. The authors used a single-layer Transformer to do the experiments and analysis. They found data replay methods succeed in shifting convergence dynamics and stabilizing pretrained knowledge. Based on this insight, the build up a new data synthesis method for replay.

**Compliance With Llm Reviewing Policy:**

Affirmed.

**Final Justification:**

I have no concerns about the paper itself after the rebuttal, though I'm still not fully convinced about how generalized the insights from the paper can be. I'll not block the acceptance if the ACs recommend to accept.

**Key Questions For Authors:**

See in weakness

**Limitations:**

yes

**Strengths And Weaknesses:**

# Strength
1. The proposed STOC method leverages attention-based token selection to guide generative replay. The connection between the theoretical insights and the algorithmic design is a positive aspect of the work.
2. The authors evaluate their method on synthetic biography datasets as well as several benchmarks for continual knowledge acquisition, showing consistent improvements over baseline replay methods.
3. The two insights about fact-to-frequency abstraction and diversity-aware attention assignment are quite interesting if they can hold for bigger models.

# Weakness
1. The theoretical analysis is interesting and internally consistent. However, it relies on a highly simplified setting (e.g., a single-layer transformer, linear attention, fixed attention scores during parts of the analysis, and structured input assumptions). These assumptions significantly deviate from modern deep transformer architectures (e.g. residual connections, layer normalization, dropout, and multi-head attention) used in large language models, which makes me confused as to why we can generalize the theoretical analysis to the actual LLMs. How does the theoretical analysis explain the grokking phenomenon in LLM's factual knowledge learning [1], and the model actually is able to generalize for factual knowledge without data augmentation [2]?


2. The paper is focused on the replay and regularization method for preventing catastrophic forgetting. Even though it's focused on the theoretical analysis, I think it's important to compare with other normal factual knowledge learning methods, like the LoRA adapters.


[1] Li, Z., Fan, C., & Zhou, T. (2025). Grokking in LLM Pretraining? Monitor Memorization-to-Generalization without Test. arXiv preprint arXiv:2506.21551.

[2] Wu, Q., Das, S., Amani, M., Ghosh, B., Khan, M. A., Gummadi, K. P., & Zafar, M. B. (2025). Rote Learning Considered Useful: Generalizing over Memorized Data in LLMs. arXiv preprint arXiv:2507.21914.

---

> ### Author Rebuttal · Authors · 2026-03-30
>
> Thanks for acknowledging our work and the constructive comments. Please kindly find point-to-point responses below.
>
> > **W1** The theoretical analysis is interesting and internally consistent. However, it relies on a highly simplified setting (e.g., a single-layer transformer, linear attention, fixed attention scores during parts of the analysis, and structured input assumptions). These assumptions significantly deviate from modern deep transformer architectures (e.g. residual connections, layer normalization, dropout, and multi-head attention) used in large language models, which makes me confused as to why we can generalize the theoretical analysis to the actual LLMs. How does the theoretical analysis explain the grokking phenomenon in LLM's factual knowledge learning, and the model actually is able to generalize for factual knowledge without data augmentation?
>
> We appreciate your concern about generalizing from single-layer theory to modern standard LLMs. We have discussed the potential effects of various components in modern Transformers, such as Softmax Multi-Head Attention and MLP, in Appendix C. As noted in the limitations section, explaining multi-layer model convergence and reconciling their differences with single-layer settings remains a significant challenge to all learning theory researchers. Instead, Section 2.3 provides empirical evidence that model-level behaviors derived from single-layer analysis are substantially aligned with those of multi-layer models.
>
> On the grokking phenomenon, we provide the following intuitive explanation grounded in our theoretical framework. Correct predictions require that the model has mastered the knowledge of all necessary tokens. However, Theorem 1 demonstrates that learning rates for different tokens vary, and token frequencies in the training corpus are heterogeneous. Due to the bottleneck effect, rapid knowledge acquisition emerges once the final, most-constrained tokens escape the initialization basin and tend to converge in the training. During the apparent plateau phase, although prediction accuracy shows non-visible improvement, partial token knowledge is already being sufficiently updated, laying the foundation for subsequent sharp accuracy gains. This explains the characteristic grokking phenomenon: a prolonged period with stagnant loss followed by sudden convergence, driven by heterogeneous convergence rates across tokens.
>
> On generalization without data augmentation, our theoretical analysis indicates that single-layer models struggle to achieve knowledge generalization without data augmentation. The fundamental difficulty is that models cannot reliably distinguish between subject/relation tokens, making it challenging to properly attribute the object token to the correct subject representation. Some existing results analyzing multi-layer settings in the Markov regime [1,2] suggest that multi-layer models may possess proper attribution capabilities. However, how these conclusions extend to knowledge learning tasks warrants further investigation. Additionally, empirical results in [3] indicate that current standard architectures are suboptimal for knowledge generalization without data augmentation. Overall, this represents an important direction for future research.
>
> [1] How transformers learn causal structure with gradient descent, ICML24.
>
> [2] Unveiling induction heads: Provable training dynamics and feature learning in transformers, NeurIPS24.
>
> [3] Physics of language models: Part 3.1, knowledge storage and extraction, ICML24.
>
> > **W2** The paper is focused on the replay and regularization method for preventing catastrophic forgetting. Even though it's focused on the theoretical analysis, I think it's important to compare with other normal factual knowledge learning methods, like the LoRA adapters.
>
> Due to space constraints in this rebuttal, we refer the reviewer to our detailed response to Reviewer TKU1 (Q2 Part), where we provide validation of some more baselines mentioned by the reviewers.  In accordance with your requirements, we fine-tuned the model using LoRA with hyperparameters set to a Rank of $128$ and an Alpha equal to $4 \times Rank$, other training hyper-parameters are set the same as other methods. Our results indicate that lower ranks hinder the acquisition of new knowledge, while higher ranks require computational costs approaching those of full-parameter fine-tuning, which contradicts the core value proposition of LoRA. Therefore, these findings demonstrate that our approach maintains the most superior performance.
>
> ---
> We sincerely thanks for the valuable comments on our paper, which help us further improve our work. We hope that our responses adequately address your concerns. If there are any further questions, please feel free to raise them. We're looking forward to further discussion.

---

> > ### Author Rebuttal · Reviewer_bpZT · 2026-04-02
> >
> > Thanks for the response. While my concern in W2 has been addressed, W1 is still not fully resolved.
> >
> > After reading the response, I would like to further clarify two major concerns regarding the insights derived from the analysis based on a one-layer transformer.
> >
> > First, regarding the architecture: as the authors note, simplifying architectures is common in theoretical analysis, which is understandable. However, my second concern, arguably more critical, relates to the role of data scaling (it actually also comes with architecture scaling). My understanding is that we study **continual** factual knowledge learning in language models precisely because they are already trained on large corpora and possess substantial prior knowledge, yet we still expect them to acquire new knowledge over time. Given that the paper’s claims center on continual factual knowledge learning, the analysis should ideally reflect a setting where the model starts from an already knowledgeable state.
> >
> > This leads to my confusion about how well the paper’s conclusions generalize. In particular, Reference [3] focuses on knowledge acquisition during the pre-training phase, with a key takeaway being that data augmentation is necessary to extract knowledge effectively later. However, this is fundamentally different from the problem of continual factual knowledge learning considered in this paper.
> >
> > As I mentioned before, I appreciate the analysis conducted on the one-layer transformer. However, I am not yet convinced that the insights can be directly extended to general language models. It may be more appropriate for the authors to narrow the scope of their claims to one-layer transformers to avoid potential overstatement of generality.

---

> > > ### Author Response · Authors · 2026-04-03
> > >
> > > Thank you for raising these thoughtful and substantive concerns. We appreciate the opportunity to provide further clarification.
> > >
> > > **On the Model's Initial State and Pre-training:** Our theoretical analysis is indeed constructed within a setting where the model begins from an already knowledgeable state. Although LLMs acquire knowledge during the PT phase, the knowledge is inherited solely through parameter initialization and thus remains vulnerable to catastrophic forgetting during CPT. In our theoretical framework, the loss landscape is convex with respect to the parameters (convexity follows from our different learning rate assumptions), which implies that the convergence point is independent of initialization conditions. At Line 271, we provide explicit analysis of how initialization conditions affect convergence rates, later supported by empirical validation. Furthermore, our proposed generative replay method is grounded in the assumption that the model has undergone adequate pre-training, a premise verified through empirical results in our real-world experiments. Therefore, we maintain that our conclusions appropriately reflect the continual factual knowledge learning setting.
> > >
> > > **On Data Augmentation and Generalization:** While our analysis elucidates how data augmentation enhances generalization, characterizing model generalization without data augmentation is not the primary focus of this work. We acknowledge relevant prior work that may provide intuitive insights into this question, however, a systematic investigation currently lacks comprehensive treatment in the literature. A rigorous investigation of this question during rebuttal presents tractability challenges, so we can only offer preliminary observation and defer it to future work. We argue that incomplete investigation of this auxiliary question does not diminish the core contributions of this paper, as we have provided substantial theoretical and empirical support for our primary results.
> > >
> > > **On Scope and Generalizability Claims:** We sincerely appreciate your constructive guidance. We commit to more explicitly emphasizing that our theoretical results are derived specifically for the one-layer transformer architecture. This clarification will be incorporated into the paper promptly to avoid overstatement of generalizability beyond the scope of our theoretical guarantees.

---

### Official Review · Reviewer_TKU1 · 2026-03-13

**Soundness:** 3
**Presentation:** 2
**Significance:** 3
**Originality:** 3
**Overall Recommendation:** 5
**Confidence:** 2

**Summary:**

This paper works to build a theoretical framework explaining why continual pre-training approaches to continual factual knowledge acquisition can incorporate new knowledge without exhibiting catastrophic forgetting. The authors first establish through mathematical proofs that facts are stored as frequency-based token contributions and that attention scores are assigned according to the diversity of relations associated with a given token. They then propose a Biography dataset as a simple synthetic benchmark for assessing continual learning, which consists of individual biographies generated through pre-defined templates to which models are prompted to fill certain attributes. Results from evaluating two small LMs on this dataset motivate their further conclusions that data augmentation alters the diversity index of tokens, pushing the model to rely on subject information for prediction, and that regularization slows forgetting but does not mitigate it entirely.

Following their previous conclusions, the authors propose Selecting Tokens via attentiOn Contribution (STOC), a generative replay method that uses token-level attention scores to select context segments for the later generation of replay data. They then evaluate this replay approach on both their own synthetic Biography dataset and several real-world benchmarks, such as KnowEdit, showing significant improvements in performance over a previously established generative replay approach, LAMOL.

**Compliance With Llm Reviewing Policy:**

Affirmed.

**Final Justification:**

The authors have conducted further experiments and included an additional architecture for comparison (Lora), which are welcome additions that strengthen the presentation of STOC. Furthermore, I believe the authors have adequately addressed the concerns surrounding the generalizability of their theoretical analysis.

**Key Questions For Authors:**

1. Unless I'm missing something, it seems that there are missing details to the STOC implementation that are also not present in the Appendix section referenced. What are the hyperparameters involved here? It is mentioned that a data selection process can filter out low-quality data, but what form would this take and is any filtering performed here? More details on this would be nice.
2. Is there another method of cFKA that you could readily include for the comparison with STOC? The evaluation seems limited with only a single comparison point (LAMOL).

**Limitations:**

Yes.

**Strengths And Weaknesses:**

**Soundness:**

The theoretical conclusions all appear technically sound based on the assumptions given, and the conclusions on regularization and data replay follow from the results of evaluating on the synthetic Biography dataset. Based on their previous theoretical assertions, the STOC replay method seems well motivated and the experimental results show that it exhibits a significant increase in performance over LAMOL in continual learning tasks.

**Presentation:**

The paper is generally well-organized, with each section logically flowing into the next, and the authors clearly indicate the conclusions drawn from each theoretical analysis. It would be nice if more details on the STOC method could be included in the main paper rather than relegating them to the appendix given that this replay method is a major contribution of the main text.

**Significance:**

The work presented here helps advance the understanding of why some continual learning approaches may perform better and/or have different behavioral impacts on downstream task performance. Although the results here are largely verified with a synthetic benchmark, the authors recognize this and should be able to extend this analysis to real-world benchmarks as well. The STOC replay method here could also have wider implications as the underlying approach is tested more thoroughly and iterated upon.

**Originality:**

The paper uses a theoretical approach to provide new insights into why data replay is an effective means of continual learning, and the authors leverage these insights to propose a novel generative replay technique that improves upon prior methods.

---

> ### Author Rebuttal · Authors · 2026-03-30
>
> Thanks for your constructive comments. Please find point-to-point responses below.
>
> > **Q1** It seems that there are missing details to the STOC implementation. What are the hyperparameters involved here? What form would data selection take?
>
> Thanks for the important questions. The previous hyperparameters involved in the STOC method include replay_data_ratio, freezing_layers, and other training hyperparameters like learning schedule. Our primary data selection strategy employs MinHash-based deduplication to filter out redundant data.
> To be mentioned, case studies reveal that the deduplication operation has a more pronounced effect on LAMOL than STOC. This is because LAMOL relies on limited prompts for generating replay data, making it more susceptible to producing highly similar samples. In contrast, STOC leverages a more diverse set of prompts, which naturally reduces the likelihood of generating redundant data. We will provide more comprehensive implementation details in the subsequent update.
>
> > **Q2** Is there another method of cFKA that you could readily include for the comparison with STOC?
>
> On Model-based vs. Data-based Methods: Other CPT methods like freezing, are model-centric methods and orthogonal to data-centric methods. Therefore, we don't treat them as direct comparison baselines but leverage them as complementary hyperparameters to be combined with. Following the suggestion from Reviewer bpZT, we also employ rank-128 LoRA as an alternative configuration for comparison.
>
> On Generative Data Replay Baselines: To the best of our knowledge, LAMOL is currently the only generative data replay baseline readily applicable. We compare against storage-based data replay on synthetic datasets. On real-world datasets the lack of available storage data prevents such comparisons. Inspired by Reviewer snZb's feedback, we replace the attention score-based token selection with random sampling, which serves as an ablation study and a baseline. We welcome any suggestions for further baseline comparisons.
>
> As shown in the following table, STOC consistently demonstrates superior performance compared to other replay data generation strategies on synthetic datasets (with model-side hyperparameters held constant). These results indicate that the newly introduced baselines and model-side training techniques do not alter STOC's superiority. Due to rebuttal time constraints, our have not yet extend the experiments to real-world settings. We will provide additional results later within the scope of our resources.
>
> | Target | Replay | hFTA | sFTA | 0.5 EM | hFTA | sFTA | 0.67 EM | hFTA | sFTA | 0.8 EM | hFTA | sFTA | 0.9 EM |
> | :--- | :--- | :--- | :--- | :--- | :--- | :--- | :--- | :--- | :--- | :--- | :--- | :--- | :--- |
> | **Continual** | Random-Full | 93.62 | 90.37 | 72.63 | 94.14 | 91.37 | 78.80 | 94.10 | 91.55 | 78.36 | 93.87 | 90.61 | 72.01 |
> | | Random-LoRA | 81.61 | 77.30 | 54.57 | 91.96 | 83.24 | 57.51 | 93.53 | 87.85 | 64.06 | 94.14 | 89.78 | 68.22 |
> | | Random-Freeze | 94.03 | 90.08 | 70.54 | 94.26 | 91.67 | 74.89 | 94.28 | 91.86 | 75.40 | 94.31 | 91.64 | 73.19 |
> | | LAMOL-Full | 94.19 | 92.58 | 85.33 | 94.35 | 92.94 | 86.84 | 94.38 | 93.13 | 87.29 | 94.08 | 92.67 | 79.76 |
> | | LAMOL-LoRA | 90.05 | 88.78 | 70.14 | 92.83 | 87.61 | 76.99 | 93.31 | 88.81 | 69.68 | 93.61 | 89.68 | 66.96 |
> | | LAMOL-Freeze | 94.34 | 92.47 | 82.50 | 94.43 | 92.62 | 81.01 | 94.51 | 93.13 | 78.56 | 94.33 | 92.06 | 75.03 |
> | | STOC-Full | 93.65 | 90.47 | 71.41 | 94.11 | 91.48 | 76.62 | 94.15 | 91.46 | 77.04 | 94.17 | 92.13 | 77.43 |
> | | STOC-LoRA | 88.19 | 79.30 | 59.98 | 91.91 | 82.22 | 63.72 | 93.64 | 87.26 | 65.33 | 94.26 | 90.08 | 69.34 |
> | | STOC-Freeze | 93.92 | 90.29 | 70.65 | 94.40 | 92.04 | 74.24 | 94.33 | 92.11 | 75.73 | 94.43 | 91.96 | 74.93 |
> | **Original** | Random-Full | 18.96 | 17.68 | 3.14 | 18.29 | 17.07 | 2.56 | 16.55 | 13.48 | 1.22 | 10.95 | 10.35 | 0.92 |
> | | Random-LoRA | 15.47 | 13.86 | 1.74 | 14.29 | 12.12 | 1.44 | 11.56 | 7.71 | 0.98 | 9.70 | 8.32 | 0.48 |
> | | Random-Freeze | 23.42 | 22.48 | 6.93 | 21.80 | 21.02 | 6.43 | 20.95 | 20.15 | 5.30 | 18.75 | 17.73 | 3.87 |
> | | LAMOL-Full | 20.98 | 19.90 | 5.95 | 20.92 | 19.80 | 6.01 | 20.45 | 19.29 | 5.75 | 12.89 | 12.15 | 2.97 |
> | | LAMOL-LoRA | 18.71 | 17.26 | 3.72 | 17.47 | 13.14 | 4.66 | 11.54 | 8.03 | 1.41 | 9.87 | 8.12 | 1.23 |
> | | LAMOL-Freeze | 25.39 | 22.18 | 9.29 | 23.14 | 21.69 | 9.47 | 22.78 | 21.33 | 9.22 | 20.06 | 18.88 | 7.58 |
> | | STOC-Full | 55.91 | 51.54 | 29.84 | 54.86 | 50.39 | 29.64 | 49.78 | 45.40 | 25.17 | 22.03 | 19.70 | 7.18 |
> | | STOC-LoRA | 51.36 | 44.75 | 18.20 | 49.93 | 41.76 | 18.26 | 46.96 | 40.25 | 17.84 | 15.14 | 11.52 | 1.96 |
> | | STOC-Freeze | **59.07** | **54.33** | **33.89** | **58.70** | **53.80** | **32.83** | **55.68** | **50.67** | **29.84** | **44.82** | **40.54** | **21.62** |
>
> ---
> Thanks again for the valuable comments. Hope our responses address your concerns. Please feel free to raise any further questions.

---

> > ### Author Rebuttal · Reviewer_TKU1 · 2026-04-03
> >
> > I appreciate the authors' response, especially the additional experimental results with the new Lora comparison that I believe would strengthen the work if included in the main paper. I also understand that time for these revisions is limited, so it is unlikely that further experiments on real benchmarks can be performed before the final decision. However, reviewer bpZT brings up a fair point questioning the generalizability of the theoretical results, which makes me hesitant to increase the score further.

---

> > > ### Author Response · Authors · 2026-04-04
> > >
> > > Thanks for the kind response, we are glad to see most of your concerns have been addressed. For the generalizability question raised by bpZT, we have added further discussion to clarify our main theoretical scenario and contributions, especially model's initial state and the scope of generalizability. Hope this additional clarification helps eliminate your concern. Thanks again for your valuable comments that help us polish our paper.

---

### Official Review · Reviewer_YhDt · 2026-03-17

**Soundness:** 2
**Presentation:** 2
**Significance:** 3
**Originality:** 3
**Overall Recommendation:** 4
**Confidence:** 3

**Summary:**

This paper propose the method STOC to generate replay data in continual pretraining to prevent catastrophic forgetting. STOC selects the tokens (snippets) with high attention scores and make the model generate replay data following these selected tokens. The authors get intuition from a one-layer transformer model in the factual knowledge acquisition task. They theoretically show that one-layer transformers would assign high attention scores to information-rich tokens in this task.

**Compliance With Llm Reviewing Policy:**

Affirmed.

**Final Justification:**

The authors added new experiments and addressed my questions. I decide to raise my score to weak accept.

**Key Questions For Authors:**

1. Can authors please explain why regularization methods don't change the converging point? I don't get the logic in line 322.
2. What is naive in table 3 and 4? Is it not using any replay data at all?
3. Section 2.3 feels a bit seperated from the rest of the paper. Also the boxed message in this section (line 248) is not convincing to me. Diversity index (DI) is defined in equation 3, but there is no empirical result measuring the DI of relation tokens in sec 2.3, making the boxed message speculative. Actually I don't find section 2.3 closely related to STOC.
4. Firgure 3 in appendix looks good -- why put it to appendix?
5. The only non-trivial baseline method is LAMOL. I'm not familiar with current SOTA replay data generation methods for continual learning. Is there only one baseline that is worth comparing with?

**Limitations:**

Yes.

**Strengths And Weaknesses:**

Strength:

The theoretical results back up for the intuition of the empirical method.

Weakness:

The empirical results are a bit confusing. Some of them does not seem very relevant to this paper. I'm not sure if the evaluation is comprehensive enough. Please see questions.

---

> ### Author Rebuttal · Authors · 2026-03-30
>
> Thanks for acknowledging our work and the constructive comments. Please kindly find point-to-point responses below.
> > **Q1** Can authors please explain why regularization methods don't change the converging point? I don't get the logic in line 322.
>
> Thanks for the question. The third term in Eq. (5) is the critical factor determining whether the model shifts its convergence point. Specifically, the $\tilde{u}$ factor contains $diag(w_s)$, whose smallest eigenvalue is $\lambda_{\min}(diag(w_s)) = \min_o w_{o,s}$, where $w_{o,s}$ denotes the importance score of parameter $y_{o,s}$.
> **Critical Constraint**: A shift in $y_s$ can only occur if every element of the pre-trained $u_s$ is substantially important. However, due to the finite knowledge storage capacity of the model [1], the knowledge stored in the entire matrix is bounded by dimension $D$. Consequently, not all elements of $u_s$ can be simultaneously important, making all the related original object knowledge about the token $s$ forgotten.
>
> [1] Physics of Language Models: Part 3.3, Knowledge Capacity Scaling Laws, ICLR25.
>
> > **Q2** What is naive in Table 3 and 4? Is it not using any replay data at all?
>
> We appreciate your clarification question. Indeed, the Naive baseline denotes the variant that does not incorporate any replay data. We will update additional explanations once available.
>
> > **Q3** Section 2.3 feels a bit separated from the rest of the paper. Also, the boxed message in this section (line 248) is not convincing to me. Diversity index (DI) is defined in equation 3, but there is no empirical result measuring the DI of relation tokens in sec 2.3, making the boxed message speculative. Actually I don't find section 2.3 closely related to STOC.
> > **Q4** Firgure 3 in appendix looks good -- why put it to appendix?
>
> We sincerely appreciate your insightful questions regarding the logical flow of the paper, particularly for Section 2.3. Due to space constraints during submission, we provisionally placed Figure 3 in the appendix and plan to incorporate it into the main text with additional pages if accepted. We recognize this arrangement may have obscured the significance of Section 2.3 and would like to clarify the core logic once more.
> Since our theoretical analysis requires empirical validation, we face a potential circularity issue if we were to verify these conclusions solely through the CPT phase, which later depends on the findings themselves. To circumvent this, we employ the PT phase as a "cross-validation". The PT phase isolates the initialization confounders while preserving the same underlying mechanisms as the CPT. Consequently, Section 2.3 provides empirical support for Section 2.2, wherein our attention-based analysis validates the feasibility of STOC's motivation. Thus, Section 2.3 is an integral and inseparable component of this work, rather than an isolated section.
> Regarding your question about the empirical verification of Theorem 2: We first acknowledge several technical challenges. (1) Multi-layer models in Section 2.3 compute Next Token Prediction for every position in the sequence, which introduces a gap relative to our simplified theoretical setting, making direct computation of DI computationally challenging. (2) The presence of multi-layer, multi-head attention mechanisms complicates the identification of appropriate proxies for attention scores. In our initial submission, we implicitly assumed that relation tokens exhibit higher DI than subject tokens, reasoning that templates expressing relations associate with richer linguistic expressions compared to name tokens. We recognize the lack of rigor in this approach and deeply appreciate your feedback.
> To further address your concern, we have retrained a model following the setting in Section 2.2 (Last Token Prediction) and computed the DI for each token according to Equation 3. We then draw a scatter plot and calculate the correlation coefficient, which is available at the anonymous link: https://anonymous.4open.science/r/continual_Factual_Knowledge_Acquision-63B1 . The results demonstrate a significant negative correlation (Pearson and Spearman correlations < -0.8), supporting the conclusion in Theorem 2.
>
> >  **Q5** The only non-trivial baseline method is LAMOL. I'm not familiar with current SOTA replay data generation methods for continual learning. Is there only one baseline that is worth comparing with?
>
> Due to space constraints in this rebuttal, we refer the reviewer to our detailed response to Reviewer TKU1 (Q2 Part), where we provide validation of some more baselines mentioned by the reviewers.
>
> ---
> We sincerely thanks for the valuable comments on our paper, which help us further improve our work. We hope that our responses adequately address your concerns. If there are any further questions, please feel free to raise them. We're looking forward to further discussion.

---

> > ### Author Rebuttal · Reviewer_YhDt · 2026-04-02
> >
> > Thank the authors for reponses and new experiments.
> >
> > The new experiment result is confusing. If I understand it correctly, Theorem 2 is saying if the DI of a token s is higher, than it gets a higher attention score (This point is also stated in line 194). But the experiment is suggesting the opposite -- low-DI tokens receive higher attention scores. Did I miss anything?

---

> > > ### Author Response · Authors · 2026-04-03
> > >
> > > Thank you for bringing this critical point. We appreciate the opportunity to clarify this apparent contradiction.
> > >
> > > **Clarification on Diversity Index Interpretation:** A higher Diversity Index indicates that a token is associated with a greater variety of objects, which correspondingly implies lower information content. A canonical example is the token "the", which appears across numerous contexts but carries minimal semantic information, such that $\ln \Pr(s|o)$  is negative with large absolute value. Therefore, our experimental results demonstrate that tokens with lower information content (higher DI) receive lower attention scores. This finding is entirely consistent with the experiments in Section 2.3 and the main theoretical arguments of our paper, where the model appropriately allocates lower attention weights to less informative tokens.
> > >
> > > **Correction to Line 194:** We acknowledge that line 194 contains a typographical error. The statement should read: "...as the DI is large". This typo inadvertently created the confusion regarding the relationship between DI and attention scores. We sincerely apologize for this error and any confusion it has caused. This typo will be corrected in the revised version.

---

### Decision · Program_Chairs · 2026-04-30

**Decision:**

Accept (regular)

**Comment:**

This paper focuses on understanding  Continual Factual Knowledge Acquisition (cFKA) in Large Language Models. The authors first presents a theoretical framework to describe the training dynamics of cFKA on simplified  single-layer Transformer model with linear attention. They find  that regularization-based methods  merely slow down catastrophic forgetting without eliminating it,  whereas data replay methods steer convergence toward a more stable state with improved knowledge retention. Next, the paper proposes STOC (Selecting Tokens via attentiOn Contribution), a novel  data replay strategy which uses attention scores to guide replay data generation and offers a mechanism for reconciling acquisition of new knowledge with the preservation of
the existing  knowledge. The authors conduct extensive experiments on both synthetic and real-world datasets  and demonstrate the effectiveness of STOC in mitigating catastrophic forgetting.

The reviews highlighted soundness of the theoretical framework, and the basis it provides for the proposed algorithm (STOC), and diverse set of experiments. The reviewers also expressed concerns about the generalizability of the theoretical analysis to more realistic settings/architectures, and interpretation of some of the empirical results. The authors addressed the concerns during the rebuttal, and provided more experimental results, which led to improved scores by three of the reviewers. Overall, I find that although the question of broader applicability remains only partially addressed, the paper offers enough value to merit acceptance.